# TPD: Enhancing Student Language Model Reasoning via Principle Discovery and Guidance

**Haorui Wang**[1], **Rongzhi Zhang**[1], **Yinghao Li**[1], **Lingkai Kong**[1], **Yuchen Zhuang**[1],
**Xiusi Chen**[2], **Chao Zhang**[1]
[1]College of Computing, Georgia Institute of Technology
[2]Department of Computer Science, University of California, Los Angeles
{hwang984,rongzhi.zhang,yinghaoli,lkkong,yczhuang,chaozhang}@gatech.edu,
xchen@cs.ucla.edu

## Abstract

Larger Language models (LLMs) often surpass their smaller counterparts in reasoning tasks but fall short in inference efficiency, posing the need and challenge of effectively transferring these capabilities from larger to smaller models. Existing approaches heavily rely on extensive fine-tuning data or continuous interactions with a superior teacher LLM during inference. We introduce a principle-based teacher-student framework, *Teaching via Principle Discovery* (TPD), to address these limitations. Inspired by human learning mechanisms, TPD mimics the interaction between a teacher and a student using a principle-based approach. The teacher LLM generates problem-solving instructions and corrective principles based on the student LLM's errors. These principles guide the refinement of instructions and the selection of instructive examples from a validation set. This enables the student model to learn from both the teacher's guidance and its own mistakes. Once the student model begins making inferences, TPD requires no further intervention from the teacher LLM. Through extensive experiments across eight reasoning tasks, we demonstrate the effectiveness of TPD. Compared to standard chain-of-thought prompting, TPD significantly improves the student model's performance, achieving an average improvement of 6.2%.

## 1 Introduction

Recent studies show that large language models (LLMs) can achieve impressive performance in various reasoning tasks, such as analogical (Webb et al., 2023), arithmetic (Imani et al., 2023), symbolic (Pan et al., 2023a), and commonsense reasoning (Wei et al., 2022b; Bang et al., 2023). However, a noticeable performance gap can often be observed between stronger LLMs such as GPT-4 and weaker LLMs such as GPT-3.5-turbo (Espejel et al., 2023; OpenAI, 2023; Cai et al., 2023). This disparity arises from factors such as training data size, model capacity, and the methods by which LLMs learn and encode world knowledge. While stronger LLMs exhibit superior performance, their practical application is hindered by the high costs associated with training and inference. For instance, as of this writing, the cost of using GPT-4 is over ten times higher compared to GPT-3.5-turbo. This raises the question: how can we effectively transfer the advanced reasoning capabilities of stronger LLMs to weaker ones?

Several approaches have been proposed to address this challenge. Some studies (Rajani et al., 2019; Ho et al., 2023) curate datasets for specific downstream tasks using stronger LLMs and then fine-tune weaker LLMs on these datasets to instill the necessary knowledge. However, this fine-tuning process is time-consuming, and the resulting task-specific weaker LLMs lack generalizability to other tasks. Wang & Li (2023) use an assistant language model to provide guidelines and analysis for the student model, but this approach requires constant involvement of the assistant model, which can be costly. To reduce teacher model involvement, Saha et al. (2023) request teacher intervention only when the student model

exhibits low confidence, where the confidence is computed from the token probability of the answer. Such a measure may not accurately reflect the student model's true confidence, as the token probability of the answer is influenced by the preceding output, specifically the chain-of-thought (CoT) explanations.

To address these limitations, we introduce Teaching via Principle Discovery (TPD), a principle-based teaching framework that minimizes teacher model involvement, thereby optimizing resource allocation and efficiency. TPD draws inspiration from instructional strategies observed in natural human teaching and learning processes (Rosenshine, 2012; Henderson & Harper, 2009; Metcalfe, 2017), which follows a structured *Demonstrate-Practice-Review* process: 1) *Demonstrate*: The teacher introduces a problem type and demonstrates how to solve it. 2) *Practice*: The student engages with practice questions. 3) *Review*: The teacher reviews the student's responses to identify common errors. This review process enables the teacher to extract fine-grained *corrective principles* that guide the student in rectifying errors and improving future problem-solving. TPD comprises two stages: *principle generation* and *principle exploitation*. In the principle generation stage, the teacher model generates problem-solving instructions and summarizes prin-

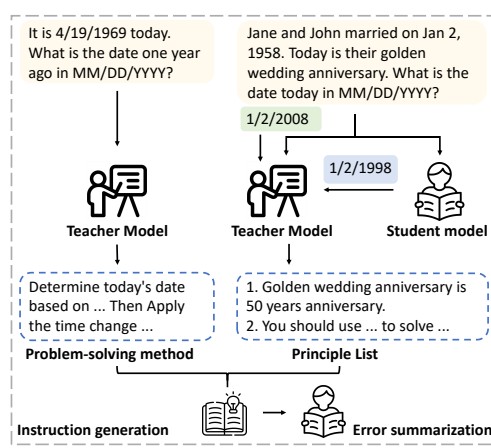

Figure 1: Illustration of TPD. The teacher model generates problem-solving instructions and then summarizes principles based on errors made by the student model on validation questions. During principle exploitation, the problem-solving instruction and examples that illustrate the principles are combined into the prompt to guide student learning.

ciples based on errors made by the student model on validation questions. In the principle exploitation stage, the teacher model constructs instructive examples that illustrate the principles and injects these instructions into the prompt to guide student learning. TPD avoids teacher model involvement during inference, enabling the student model to operate independently in offline scenarios.

We validate TPD on eight reasoning tasks covering symbolic and arithmetic reasoning. Through principle generation and exploitation, TPD significantly improves the performance of the student model without teacher model intervention during inference. Specifically, our model achieves an absolute gain of up to 19% accuracy compared to chain-of-thought prompting. Additionally, we explore different methods for injecting generated principles into the student model and find that selecting new examples from practice questions outperforms direct injection and the critique-revise method.

## 2 Related work

**Teacher-student framework**  Traditional teacher-student frameworks employ fully supervised fine-tuning methods (Magister et al., 2022; Shridhar et al., 2022; Hendrycks et al., 2021; Rajani et al., 2019). For instance, Ho et al. (2023) generate CoT reasoning steps by GPT-3 (Brown et al., 2020) and fine-tune several relatively small models with these generated data. However, both the fine-tuning process and collecting task-specific fine-tuning data are time-consuming and challenging. To overcome these limitations, there are several prompting-based teacher-student frameworks recently (Pruthi et al., 2022; Saha et al., 2023; Yu et al., 2023). In these frameworks, the teacher model is tasked with offering demonstrations or explanations to the student model. However, these approaches require the intervention of a teacher model on the test set. Hence, they are unsuitable for offline scenarios, where the student model must solve problems independently, without assistance from the teacher model.

**Eliciting LLM's reasoning ability through prompting**    Emergent abilities bring a strong few-shot learning ability for LLMs across various datasets via in-context learning (Wei et al., 2022a;b). Many prompting-based methods are proposed to elicit the reasoning abilities of the LLMs by injecting knowledge into prompts. For instance, *chain of thought* (CoT)  (Wei et al., 2022b) and its variants (Kojima et al., 2022; He et al., 2023) provide a few human written examples or instructions to LLMs. We summarize more prompting methods in Appendix B. These prompting methods are effective in eliciting the reasoning ability of the LLMs. However, they are not suitable for direct adaptation within the teacher-student framework, as they do not involve knowledge transfer from a teacher model.  Another pipeline of prompt engineering is automatic prompt searching, also known as prompt optimization (Khattab et al., 2023; Yang et al., 2023a).  These methods employ LLMs to refine prompts iteratively in the zero-shot setting for a specific task.  However, applying automatic prompt searching methods for prompt optimization is time-consuming, as it requires multiple rounds of inference on the training dataset; in contrast, TPD requires only a single-round knowledge transfer. Moreover, TPD includes an error summarization stage, which aligns with learning from feedback prompting methods.  These methods initially create a memory list. When a new query arises, they retrieve relevant information from this memory to refine and enhance the current prompt. The retrieved information may consist of successful trials from history (Majumder et al., 2023), the most similar questions with user feedback (Madaan et al., 2022a), or summaries of previous failures (Shinn et al., 2024).

**Principle discovery and exploitation**    Rule discovery is a popular technique in machine learning and data mining (Tweney et al., 1980; Fürnkranz & Kliegr, 2015; Das et al., 1998). In language models, rule discovery is often utilized in learning from feedback frameworks (Zhang et al., 2022a; Pan et al., 2023b). In (Zhu et al., 2023), rules are generated by language models and subsequently selected through self-verification. Yang et al. (2023b) propose a framework where language models could generate rules by learning from previous mistakes. However, the rules produced in these methods tend to be concrete facts, with the framework functioning as a retrieval system. In contrast, a rule is a formal, broad statement that applies to an indefinitely large set of objects. Unlike rules, principles are more abstract and open to interpretation, offering high-level guidance without strict formatting. For instance, Bai et al. (2022) predefine principles about helpfulness and harmfulness by human experts, then ask the language model to evaluate the generation results to make RLAIF. One concurrent work on principle discovery is LEAP (Zhang et al., 2024), which focuses on summarizing both high-level and low-level principles based on a model's previous mistakes. The key distinction between TPD and LEAP is that we incorporate principle discovery within a teacher-student framework, aiming to enhance knowledge transfer through these principles. In contrast, LEAP primarily demonstrates that LLMs also discover principles from their previous mistakes.

## 3    Method

### 3.1    Overview

In TPD, we consider a teacher model and a student model. The student is responsible for solving reasoning problems based on the teacher model's guidance.  The primary goal of our framework is to let the teacher model effectively teach the student model how to solve these tasks. As shown in Fig. 2, the framework is divided into two stages: **principle generation** and **principle exploitation**. During the principle generation stage, the teacher model produces a problem-solving instruction based on sampled questions and identifies a principle list $\mathbb{P}$ from the errors made by the student model. In the principle exploitation stage, the principle list $\mathbb{P}$ is injected into the student model to improve its reasoning capabilities in downstream tasks.

### 3.2    Principle generation

In this stage, we employ the teacher model to provide a high-level guideline to the student model and help the student model identify its common errors from a few practice questions.

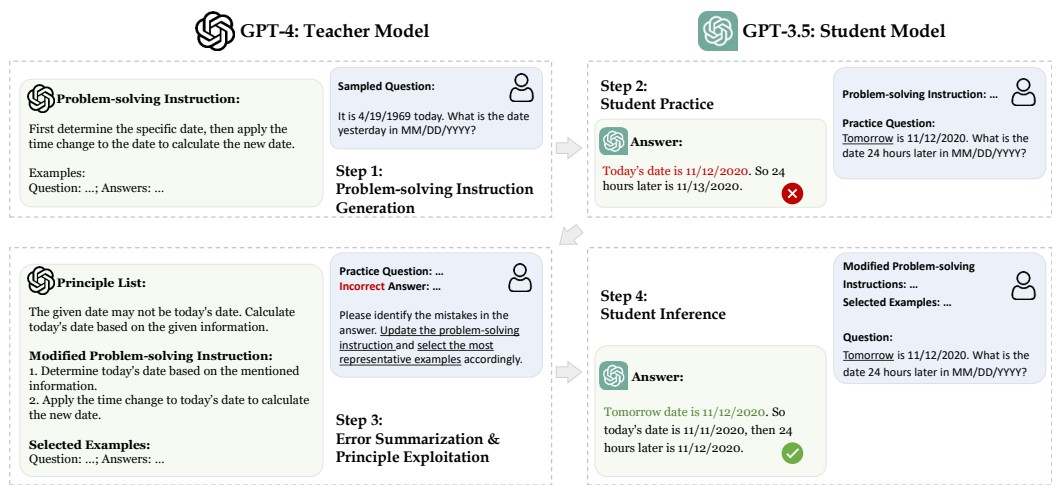

Figure 2: TPD contains two stages: principle generation and principle exploitation. In principle generation, the student model generates answers according to the problem-solving instructions from the teacher model. Then, the teacher provides a list of principles based on student's practice errors. In principle exploitation, the teacher model refines the instruction and chooses representative examples, which are used by the student for inference.

This principle generation stage comprises two sub-stages: 1. Problem-solving instruction generation and initial practice and 2. Error summarization. The two-stage process mirrors the traditional classroom learning model, where teachers provide instructions, students practice, and teachers offer feedback based on observed errors.

### 3.2.1 Problem-solving instruction generation and initial practice

**What is problem-solving instruction?** The problem-solving instruction $I$ consists of a problem-solving method augmented with a few examples. To build the instruction, the teacher model reviews a set of questions sampled from the training set, identifying what type of questions the student model needs to solve. Then, the teacher model generates a problem-solving method as an initial instruction in natural language. To enhance the instruction, it is further enriched with examples that demonstrate the application of the proposed method. These examples are derived from the same questions initially sampled. We provide an example in Appendix C for a better understanding.

**Why problem-solving instruction?** Previous works (Saha et al., 2023; Wang & Li, 2023) ask the teacher model to offer guidance for each test question, rendering the teaching framework impractical for offline scenarios. To tackle this issue, we ask the teacher model to provide high-level problem-solving instruction based on sampled questions and internal knowledge. The problem-solving instruction is adaptable, making it suitable for offline use as it generalizes across similar types of questions. Moreover, since many prompting methods (Wei et al., 2022b; Gao et al., 2023b; Zhou et al., 2022) have shown that LLMs have a strong capability to learn from examples, the problem-solving instruction includes a few examples to help the student model understand and imitate the instruction. This process also follows the *Case Teaching Method* paradigm (Herreid, 2005), where the teacher models provide several cases, and the student model can imitate and learn from the examples. The problem-solving instruction reflects the key principles to solve the type of problems, as it provides fundamental guidance for addressing specific types of problems.

**Initial practice for the student model** With the problem-solving instruction, the student model generates answers for the questions in the validation set. This step lets the student model practice, allowing the teacher model to assess its comprehension of the problem-solving instruction and its ability to apply it effectively.

### 3.2.2 Error Summarization

**Data filtering** After the student practices over the validation set, we evaluate its responses to build an error set $\mathbb{E} = \{e_1, e_2, ..., e_n\}$, where each element $e_i$ in this set represents a pair consisting of a question and the corresponding incorrect answer made by the student model. Although the teacher model generally performs well on various tasks, it can make mistakes. Hence, we need to assess the teacher model's ability to accurately identify the incorrect answers in $\mathbb{E}$. Specifically, we check if the teacher model agrees with the student's incorrect answers, removing each $e_i$ that the teacher model can not identify to build a feasible error set $\hat{\mathbb{E}}$. Due to the context length of the LLMs, we then develop an iterative framework to summarize principles from each $e_i$ iteratively.

**What is error summarization?** In Algorithm 1, we denote the model response as $M(p)$ given the LLM $M$ and the input $p$. The teacher model initially derives principles $\mathbb{P}$ from a subset $\mathbb{N}_s$ of the feasible error set $\hat{\mathbb{E}}$. Then, we present the remaining set $\mathbb{N}_r$ to the teacher model sequentially, prompting it to determine whether the established principle list $\mathbb{P}$ can rectify the error. If the existing principle list $\mathbb{P}$ can not address the presented error $\mathbb{N}_r(i)$, the teacher model will formulate a new principle $p$ for it. The iterative process stops when all the errors in $\mathbb{N}_r(i)$ are checked. Then, human reviewers will step in to assess the validity of the principle list $\mathbb{P}$ created by the teacher model. The reviewers will simply delete any that are found to be erroneous or confusing, ensuring the reliability and clarity of the final principle list. We provide instructions for human reviewers in Appendix F.

---

**Algorithm 1:** Error summarization

---

**Input:** $T$: the teacher model; $\hat{\mathbb{E}}$: the feasible error set
**Output:** $\mathbb{P}$: the principle list
1 Sample a subset $\mathbb{N}_s$ from $\hat{\mathbb{E}}$;
2 $\mathbb{N}_r \leftarrow \hat{\mathbb{E}} \backslash \mathbb{N}_s$;
3 $\mathbb{P} \leftarrow T_{\text{summarize}}(\mathbb{N}_s)$;
4 **for** $i \in \{1, 2, 3, \ldots, |\mathbb{N}_r|\}$ **do**
5    **if** $T_{\text{evaluate}}(\mathbb{P}, \mathbb{N}_r(i)) = \text{False}$ **then**
6       $p \leftarrow T_{\text{summarize}}(\mathbb{N}_r(i))$;
7       $\mathbb{P} \leftarrow \mathbb{P} \cup p$;
8    **end**
9 **end**
10 **return** $\mathbb{P}$

---

**Why error summarization?** Making mistakes is a natural part of the learning process and can be a powerful tool for growth and understanding (Cyr & Anderson, 2018). Even though students can solve simple tasks using the original problem-solving instruction alone, they may have difficulty applying the same knowledge and skills in complex situations (Klein et al., 2007). Therefore, it is also important to let students learn from errors, which can help them identify error patterns and avoid similar mistakes in future practice (Metcalfe, 2017). Learning from mistakes has been explored in (Wang & Li, 2023; Yang et al., 2023b), where knowledge from past failures is stored in a memory list and retrieved for each new query. However, these approaches primarily focus on recording errors in detail without summarizing them into high-level principles. In contrast, our approach emphasizes high-level principles, which are more generalizable and can be effectively applied in offline scenarios. Additionally, TPD can also be augmented by retrieval methods, as we can retrieve the most similar questions along with the teacher model's feedback from practice questions. Further analysis is provided in Appendix D.

### 3.3 Principle exploitation

In the second stage, the student model's role is to utilize the principle list $\mathbb{P}$ to solve various instances of the task. There are multiple ways to utilize the principle list, which are explored in Sec 4.4. Compared to directly injecting the principle list into the prompt and the critique-revise method, we found that using the principle list $\mathbb{P}$ to curate new examples from the validation set is more effective. Specifically, we define the violation score, which is the number of violations against the principles in the list $\mathbb{P}$. The teacher model evaluates each error in $\hat{\mathbb{E}}$, using violation scores to rank them. The examples with the top $k$ highest violation scores are selected as the most informative examples. Then, the teacher model generates correct answers for these examples to replace the incorrect answers from the

student model. Besides, the teacher model also revises the problem-solving instruction based on $\mathbb{P}$. The overall instruction combines the revised problem-solving instruction and the selected informative examples. For better illustration, we present the overall instruction template below.

---
**Overall Instruction Prompt**

Revised Problem-solving Instruction: {*method*} + {*original examples*}
New Selected Examples: {*question*} + {*answer*}

---

The principle generation stage only needs to be performed once for each type of task. The teacher model could then build a new prompt based on the principle list for the student model. The prompt can then be reused for all instances of that task. This makes TPD significantly more efficient than using a retriever or a teacher model's intervention in each query.

## 4 Experiments

### 4.1 Experiments Setup

**Datasets.** We evaluate our approach on eight datasets from diverse domains, including four tasks from Big-bench (Srivastava et al., 2022): Tracking Shuffled Objects, Date Understanding, Navigate, and Matrixshapes. The other four datasets are GSM8K (Cobbe et al., 2021), SVAMP (Patel et al., 2021), CoinFlip and Last Letter Concatenation (Kojima et al., 2022; Wei et al., 2022b). We use the 5-object versions of the Tracking Shuffled Objects task in the experiments, which we will refer to as Tracking Shuffled Objects (5). The CoinFlip, Last Letter Concatenation and Tracking Shuffled Objects (5) are regarded as symbolic reasoning tasks, and the remaining datasets are arithmetic reasoning tasks. The dataset split and the detailed information about each dataset can be found in Appendix G.

**Experiment settings.** In the experiment, we use gpt-3.5-turbo-16k, LLama3-70B-Instruct (AI@Meta, 2024) and Mixtral-8x7B-Instruct (Jiang et al., 2024) as student models, with gpt-4 (OpenAI, 2023) serving as the teacher model. In the following sections, we will denote them as GPT3.5, LLama3, Mixtral, and GPT4, respectively. More detailed settings are given in H.2.

**Baselines.** We include Zero-Shot CoT (Kojima et al., 2022), few-shot CoT (Wei et al., 2022b), and Auto-CoT (Zhang et al., 2022b) as baseline prompting methods. Specifically, we include 6-shot CoT as a baseline method, where all of the 6 examples are selected based on the principle list from practice questions (there are 6 examples in the final prompt for the student model). In contrast, 3-shot-CoT uses randomly selected examples from the training set. More details of the baselines are given in H.3.

### 4.2 Symbolic reasoning

Symbolic reasoning entails the use of symbols and their relationships to execute logical operations (MacColl, 1897). It gauges logical reasoning and rule-based decision-making abilities, assessing a language model's proficiency in simulating human-like reasoning. Table 1 presents the comparison results. As anticipated, 3-shot CoT and Auto-CoT surpass zero-shot CoT by providing more examples and guidance to the language model, enabling it to generate higher-quality reasoning processes. Auto CoT achieves comparable performance across all three datasets compared to 3-shot CoT, suggesting that it can match the performance of the CoT paradigm that requires manual designs. Our method TPD outperforms 3-shot CoT on CoinFlip and Last Letter Concatenation, and achieves comparable performance on Tracking Shuffled Objects (5), demonstrating the potential effectiveness of problem-solving instructions over merely presenting a sequence of language reasoning steps.

The error summarization stage appears to be less effective in symbolic reasoning tasks, primarily because errors often stem from a lack of factual knowledge rather than the misapplication of principles. Principles serve as high-level guidelines, applicable to a broad range

| Model | Method | Symbolic reasoning | | | Arithmetic reasoning | | | | |
|---|---|---|---|---|---|---|---|---|---|
| | | Coin | Letter | Shuffled | Date | Navi. | GSM8K | Matrix | SVAMP |
| GPT3.5 | 0-shot CoT/PoT | 65.5 | 48.9 | 47.5 | 24.0 | 39.0 | 66.7 | 26.5 | 77.6 |
| | 3-shot CoT/PoT | 80.8 | 82.9 | 74.5 | 68.5 | 81.5 | 74.5 | 84.5 | 82.6 |
| | Auto CoT/PoT | 80.6 | 81.5 | **75.5** | 71.0 | 83.0 | 73.9 | 83.5 | 82.8 |
| | 6-shot CoT/PoT | 92.8 | 83.5 | 75.0 | 74.0 | 93.0 | 74.8 | 89.5 | 82.8 |
| | TPD w/o ES | **100.0** | 89.7 | 75.0 | 33.5 | 85.0 | 74.7 | 85.0 | 81.2 |
| | TPD w/ ES | **100.0** | **89.9** | 75.0 | **76.5** | **97.5** | **75.4** | **93.5** | **82.9** |
| Mixtral | 0-shot CoT/PoT | 45.6 | 15.2 | 24.5 | 18.0 | 47.0 | 51.4 | 8.5 | 69.9 |
| | 3-shot CoT/PoT | 78.4 | 58.7 | 42.0 | 35.5 | 72.5 | 57.2 | 64.5 | 79.4 |
| | Auto CoT/PoT | 76.5 | 58.4 | 43.5 | 39.5 | 73.0 | 56.8 | 71.0 | 79.6 |
| | 6-shot CoT/PoT | **78.8** | 58.3 | 48.5 | 44.5 | 82.0 | 61.2 | 84.5 | 79.8 |
| | TPD w/o ES | 67.2 | 49.3 | 51.5 | 54.0 | 76.0 | 63.6 | 68.0 | 84.2 |
| | TPD w/ ES | **78.8** | **59.2** | **52.5** | **71.5** | **96.0** | **65.1** | **91.0** | **86.0** |
| LLama3 | 0-shot CoT/PoT | **100.0** | 82.2 | 63.0 | 31.0 | 83.0 | 81.3 | 61.0 | 82.5 |
| | 3-shot CoT/PoT | 97.6 | 85.6 | **99.0** | 52.0 | 58.5 | 87.5 | 86.5 | 91.2 |
| | Auto CoT/PoT | 97.8 | **86.2** | 98.5 | 59.5 | 80.5 | 89.8 | 89.5 | 90.8 |
| | 6-shot CoT/PoT | 97.9 | 85.8 | **99.0** | 75.0 | 89.5 | 88.3 | 94.0 | 91.5 |
| | TPD w/o ES | 94.6 | 84.7 | 98.0 | 74.5 | 68.5 | 87.7 | 82.5 | 89.7 |
| | TPD w/ ES | 99.1 | 86.1 | **99.0** | **85.5** | **93.0** | **92.7** | **97.5** | **93.4** |

Table 1: Performance on symbolic reasoning and arithmetic tasks, measured in accuracy(%). The teacher model is GPT4, and the student models are GPT3.5, Mixtral and LLama3. In symbolic reasoning tasks, we utilize CoT as our base prompting method. In arithmetic reasoning tasks, we utilize PoT as our base prompting method. We test the performance with and without error summarization (ES).

of scenarios. They do not change with specific situations but offer a consistent approach to problem-solving. In contrast, factual knowledge, which entails detailed, verifiable information about the world, is crucial in addressing specific cases. It should be incorporated into the model's parameters during training or sourced from external databases when needed. For instance, consider the Last Letter Concatenation task: the student model adheres to the problem-solving instruction to identify and concatenate the last letters of words to form a new string. The principle guides the process, but errors may occur due to inadequate factual knowledge about the last letter of a particular word within the string.

## 4.3 Arithmetic reasoning

The results of arithmetic tasks are shown in Table 1. Zero-shot PoT results in the worst performance in all datasets. The main issue for 0-shot PoT is the LLMs' inability to utilize their pre-trained knowledge to craft solutions for problems without explicit guidance, thereby failing to tap into their potential for coding and logical reasoning with a basic prompt alone. The Auto-PoT approach, which stratifies validation by setting questions into three clusters and selecting the example closest to the center from each cluster to form a 3-shot PoT, aims to introduce diversity into the examples. However, it only marginally improves performance across three datasets and performs poorly on GSM8K, which suggests that a mere variety in questions does not inherently lead to distinct reasoning pathways. The original problem-solving instruction derived from the teacher model demonstrates performance comparable with, and occasionally inferior to the 3-shot PoT, since manually created examples may surpass those generated by LLMs in quality.

In comparison, TPD with error summarization outperforms other prompts significantly over Date Understanding, Navigate, and Matrixshape tasks, which indicates that a principle list, distilled from the analysis of errors in practice questions, is highly effective. It guides the teacher model in refining the original problem-solving instruction and choosing the most informative samples for few-shot prompts. However, it only achieves comparable performance with 3-shot PoT on GSM8K and SVAMP, since the errors made by the student model are diverse understanding errors caused by lack of factual knowledge and can not

be well categorized in a high-level principle list. Moreover, 6-shot-PoT outperforms other baselines, suggesting that examples selected based on the principle list are more informative than those chosen randomly from the training set.

## 4.4 How to utilize the principle list

With the principles identified by the teacher model, we explored methods to transfer this knowledge to the student model effectively. One straightforward approach involves directly appending the list of principles to the prompt. Alternatively, we can employ an iterative learning process, where the student model initially attempts to answer a question, then provides feedback based on its own understanding of the principles, and then revises its initial response. This critique-and-revise strategy has been successfully used in various

| Method | Date | Nav. | GSM8K | Matrix | SVAMP |
|---|---|---|---|---|---|
| **No principle** | 68.5 | 81.5 | 74.5 | 84.5 | 82.6 |
| **Injecting Into Prompt** | 71.5 | 83.5 | 74.8 | 86.5 | 82.6 |
| **Critique + Revise** | 61.5 | 74.5 | 71.4 | 79.5 | 74.8 |
| **Examples Selection** | **76.5** | **97.5** | **75.4** | **93.5** | **82.9** |

Table 2: Performance of different principle injection methods on GPT3.5 on arithmetic reasoning tasks, measured in accuracy(%).

prompting methods (e.g., (Chen et al., 2023; Bai et al., 2022; Madaan et al., 2023)). Our proposed method enhances this approach by selecting highly informative examples from the validation set, guided by the principle list. This strategy capitalizes on the emerging capabilities of LLM, enabling the student model to learn effectively from exemplary instances.

The experimental results presented in Table 2 reveal that simply adding a principles list to the prompt yields only a marginal improvement over the base prompt. This suggests that the student model faces challenges in effectively using high-level and implicit principles expressed in natural language. While retrieval methods gather specific factual knowledge and integrate it with the initial prompt, the principles here are more abstract and harder for LLMs to lever-age. For instance, a principle might serve

| Model | Error | GSM8K | Date |
|---|---|---|---|
| **Gpt3.5** | $e_1$ | 85% | 17% |
| | $e_2$ | 15% | 47% |
| | $e_3$ | 0% | 36% |
| **Mixtral** | $e_1$ | 78% | 16% |
| | $e_2$ | 22% | 58% |
| | $e_3$ | 0% | 26% |

Table 3: Percentage of different error types in TPD.

as a directional guide for a specific step in the reasoning process. Additionally, LLMs tend to lose information in longer contexts (Liu et al., 2023), which could further contribute to the limited impact of the principles list. Surprisingly, the critique-and-revise method resulted in a decrease in performance across all datasets. Our observations indicate that when prompted to provide feedback or critique based on the list of principles, the LLM tends to perceive the original answer as incorrect and significantly overhauls it. This behavior might stem from the RLHF stage, where terms like *feedback* or *critique* could trigger the model to question its previous outputs.

Using the principle lists to select examples from the validation set for in-context learning achieves better performance across all datasets than the other two methods. The chosen examples based on the principle list contain the most error-prone questions for the student model, thus helping the student model learn from errors effectively. This also mirrors real-world classroom dynamics, where providing students with practical examples often proves more beneficial than solely relying on textbook knowledge (Shafto et al., 2014).

## 4.5 Error Analysis

Despite the guidance provided by the teacher model, the student model may still make errors. In this section, we analyze the sources of these errors within the TPD framework. Firstly, it is important to note that the teacher model has its own limitations, especially in

complex reasoning tasks (Huang et al., 2023). As a result, it cannot detect all the errors made by the student model. To address this, data filtering is necessary during the error summarization stage. During the principle exploitation stage, errors can be categorized into three types:

- $e_1$: The principle provided is unhelpful.
- $e_2$: The student model fails to answer correctly even with a helpful principle.
- $e_3$: There are no relevant principles in the principle list for the given question.

We analyzed the errors made by GPT-3.5 and Mixtral on the GSM8K and Date Understanding tasks, with the results shown in Table 3. In the GSM8K task, no $e_3$ errors were observed for either student model, as the teacher model provided highly general principles (e.g., "Ensure arithmetic operations are logically sound and mathematically correct"). While these general principles are broadly applicable, they fall short when addressing specific questions. This indicates that an effective principle should target a particular type of error rather than rely on vague or general terms. In contrast, for the Mixtral model, $e_2$ errors were predominant. Even though the principle was helpful, the student model still failed to answer correctly, highlighting the inherent limitations of the student model's capabilities. Regarding $e_3$ errors, these arise when practice questions do not cover all possible question types, leading to gaps in the principle list. This is a common limitation of learning from feedback frameworks.

## 4.6 Ablation study

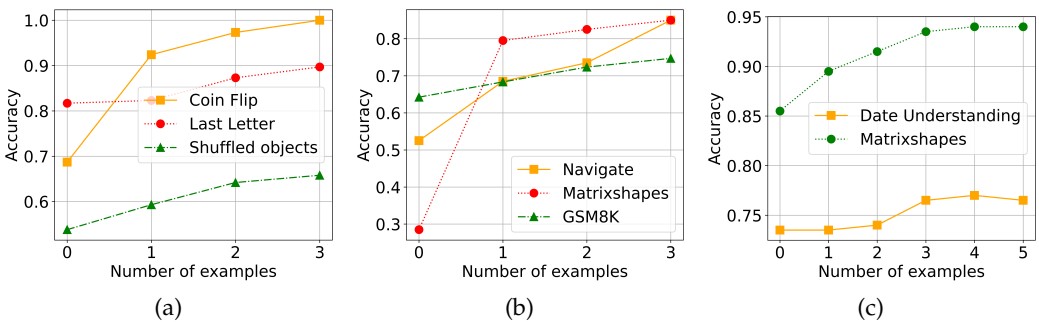

(a)  (b)  (c)

Figure 3: The test accuracy of different numbers of examples in the problem-solving instruction in (a) symbolic reasoning tasks and (b) arithmetic reasoning tasks. (c) An ablation study on the number of examples selected based on the principle list. 0 example means the prompt only contains the modified problem-solving instruction. The experiments are conducted on GPT3.5.

**Effectiveness of examples in the problem-solving instruction.** We investigate the effectiveness of having the teacher model simply describe a problem-solving method in the instruction without including examples. The results are shown in Fig. 3a and Fig. 3b. There is a significant decrease in accuracy for both tasks when examples are omitted from the problem-solving instruction, indicating the importance of incorporating examples in the problem-solving instruction. In more complex tasks like Matrixshapes and Navigate, we observed a notable increase in performance with the inclusion of the first example, while subsequent examples contributed to less. This demonstrates that for complex tasks, the student model struggles to learn from descriptions alone and relies on concrete examples to comprehend the problem-solving instruction. Across all datasets, more examples will help more, but the performance gain will be trivial as the number of examples increases.

| Model | GPT3.5 principles | | Mixtral principles | |
|---|---|---|---|---|
| | Date | Navigation | Date | Navigation |
| **GPT3.5** | 76.5 | 97.5 | 73.0 | 91.5 |
| **Mixtral** | 68.5 | 96.0 | 71.5 | 96.0 |

Table 4: Performance of student models with different sources of principles.

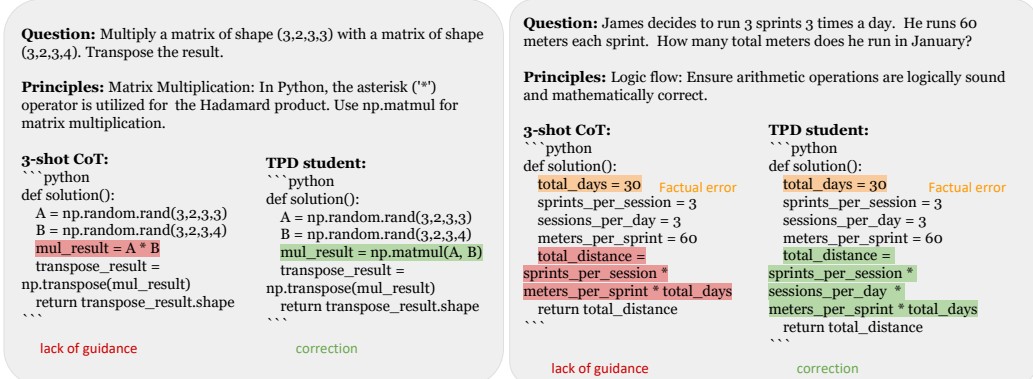

Figure 4: The case study of TPD.

**Generalization capability of principles.** We further examine if the principles derived from the errors of one student model can be effectively applied to another. As shown in Table. 4, both student models achieve good performance when leveraging principles deduced from the errors of the counterpart model. However, a slight performance drop is observed when a model utilizes principles from the other model compared to its own. This gap demonstrates that although there exist common mistakes between the two student models, there are also some differences in their error patterns.

**Numbers of examples selected in principle exploitation stage.** To determine the minimum number of examples required for effective learning, we conduct experiments with different numbers of examples selected based on the violation score. The results are presented in Fig. 3c. Our findings reveal that the initial example provides the most significant performance improvement across both datasets. This suggests that the initial example serves as the most instructive instance for the student model. The minimum number of examples required varies across datasets. For Matrixshapes, three examples are sufficient to achieve stable performance. However, while three examples provide a substantial gain for Date Understanding, adding more examples continues to benefit the student model. For simplicity and consistency across all datasets, we employ three examples in our experiments.

## 4.7 Case study

In Fig. 4, we present a comparative case study to show the success mode (left subfigure) and the failure mode (right subfigure) of TPD. In both scenarios, the principles guide the student model towards reasoning more effectively. This guidance becomes particularly evident in the success mode, where it significantly enhances the student's problem-solving capabilities. However, the principles have their limitations, especially when confronted with the model's gaps in factual knowledge. For instance, as illustrated in the right subfigure, the student model fails to accurately assign 31 days to January.

## 5 Conclusion

In this paper, we present a framework named Teaching via Principle Discovery (TPD). This approach empowers teacher models to construct problem-solving instructions and summarize key principles by analyzing example questions and student errors. The identified principles are subsequently employed to refine the problem-solving instructions and select the most informative examples from the validation set to create a tailored instruction prompt. We validate the effectiveness of TPD on symbolic and arithmetic reasoning tasks, observing a marked improvement in the performance of the student model. Our method introduces an innovative approach, utilizing advanced LLMs to guide weaker agents in tackling reasoning problems. In the future, we plan to study how to apply TPD to solve complex reasoning tasks, such as serving as a web agent, where the principle list will be much longer.

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

# A    Limitations

One limitation of the TPD is its inability to rectify common sense errors (lack of factual knowledge), as evidenced by its marginal improvements in the GSM8K and SVAMP datasets in Table 1, despite the teacher model's high accuracy (97%, Zhou et al., 2023). This limitation arises because the errors made by the student model predominantly relate to factual knowledge, a category not effectively addressed by the generated principles and instructions. High-level principles can resolve only a limited number of such errors. A potential solution for this challenge could involve fine-tuning the LLM to internalize a broader range of common sense knowledge. Another constraint of TPD is the method for applying principles. While selecting examples based on validation scores has proven more effective than other strategies, this approach is not particularly efficient. The issue stems from the LLMs' context length limitations. This inefficiency poses an unresolved challenge: effectively integrating a long list of principles into LLMs. Moreover, we do not check the overlap of the principles in the principle list, hence there might be near-duplicate principles in the principle list.

# B    Additional related work

 Madaan et al. (2022b); Gao et al. (2023a); Chen et al. (2022) find that using LLMs to generate codes for reasoning tasks and then utilizing the codes by program interpreters to solve the questions can achieve better performance. Another prompting method is the problem decomposition  (Zhou et al., 2022; Drozdov et al., 2022; Dua et al., 2022), which asks LLMs to decompose the tasks into several subtasks and solve them individually. Yao et al. (2023) propose *Tree of Thoughts*, which enables language models to self-evaluate intermediate "thoughts" and decide whether to explore different ideas or reevaluate when needed to provide the optimal solution. Self-refinement methods  (Madaan et al., 2023) ask LLMs to refine their original answers iteratively but will rely on the handwritten few shot examples.

# C    Problem-solving instruction example

The problem-solving instruction consists of a problem-solving method augmented with a few examples. These examples are derived from the same questions initially sampled. Here is an example.

> **Problem-solving Instruction:**
>
> To solve the problem where you are asked to take the last letters of each word in a given string and concatenate them, you can follow these steps:
> 1. Identify and list each word in the string.
> 2. Locate the last letter of each word.
> 3. Concatenate these letters to form a new string. Provide the new string as the answer.
> **Examples:** Question: Take the last letters of each word in "Jeremiah Kelley Josue Veronica" and concatenate them.
> Identify the words: Jeremiah, Kelley, Josue, Veronica.
> Last letters: h, y, e, a.
> Concatenate: "hyea".
> Answer: hyea.

# D    TPD with retrieval augmentation

TPD leverages the teacher model to extract and summarize principles from the mistakes made by the student model on practice questions. These principles operate at the task level. In contrast, popular learning-from-feedback frameworks typically provide feedback at the query level, where a retriever fetches relevant information for each query from a memory module. This contrast led us to explore whether TPD could be further enhanced by integrating a retrieval module. Specifically, we instructed the teacher model to generate a

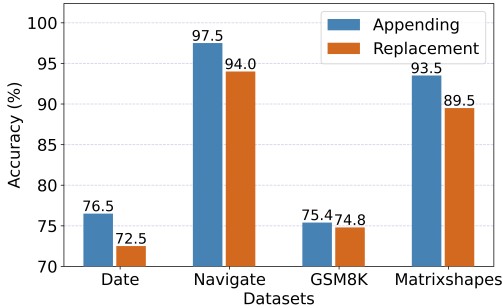

Figure 5: Ablation studies on how to utilize selected examples with the modified problem-solving instruction.

textual analysis for each error within the feasible error set $\hat{\mathbb{E}}$. During the student's inference stage, a retriever identifies the most similar question from $\hat{\mathbb{E}}$ and appends the retrieved question along with the teacher model's analysis to the student's prompt.

The results shown in Table 5 suggest that TPD can indeed be improved with the addition of a retrieval module. This improvement indicates that the task-level principles and the directly retrieved examples are complementary to some extent. However, in the Matrix task, the Mixtral model's performance did not benefit from the retrieval module. We believe that finding an optimal method to combine task-level principles with query-level retrieved information could be a promising future direction for the teacher-student framework.

|  | model | Date | Matrix |
|---|---|---|---|
| **TPD** | **GPT3.5** | 76.5 | 93.5 |
| | **Mixtral** | 71.5 | 91.0 |
| **TPD** | **GPT3.5** | 81.0 | 94.0 |
| **+RAG** | **Mixtral** | 78.5 | 91.0 |

Table 5: Performance of TPD and TPD augmented with RAG (retrieval).

## E  Additional ablation study

**Can we replace the original examples in the problem-solving instruction with the newly selected samples?**  Fig. 3c demonstrates the high informativeness of examples selected based on violation scores. This raises an intriguing question: can we directly replace the original examples in the modified problem-solving instruction with these newly chosen ones? To explore this, Fig. 5 presents a comparison between two approaches: appending the new selections to the original examples versus completely replacing them. Interestingly, the appending method consistently outperforms the replacement approach across all datasets. This superior performance can be attributed to the fact that the new selections specifically address the errors encountered when the student model learns from the original examples. Moreover, completely removing the original examples may inadvertently shift the student model's errors, rendering the newly selected examples less informative.

**Methods description.**  Our study also investigates whether the teacher model should offer a high-level reasoning method in problem-solving instruction. As illustrated in Fig. 6, we observe a notable decline in problem-solving performance without an explicit method description across three datasets. This suggests that while the student model can implicitly learn and mimic reasoning from examples, it remains crucial for the teacher model to provide a problem-solving method in natural language explicitly.

## F  Human intervention instructions

Regarding human intervention for quality checks on principles, since the principle lists in our experiments are not very long, human reviewers can finish checking the principle list

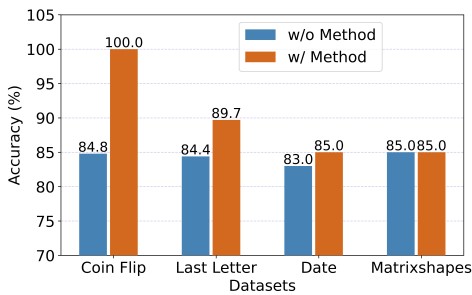

Figure 6: Ablation studies on whether the teacher model needs to provide problem-solving methods in the problem-solving instruction.

within approximately 20 minutes per task. In TPD, human reviewers are responsible for verifying each principle in the principle list for all tasks to ensure quality. They sequentially review and remove any principles that are vague, erroneous, or confusing.

> Human intervention instructions:
>
> 1.The principle should be written in simple, clear language that is easy to understand.
> 2.The principle should be specific enough to address a particular type of error or concept.
> 3.The principle should be factually correct and free from errors.
> 4.The principle should provide actionable guidance that helps the student avoid the error in the future.
> 5.While specific, the principle should also be general enough to apply to similar errors or situations.

## G   Datasets

The **Coin Flip** task (Wei et al., 2022b) requires the LLM to determine if a coin remains heads up following a series of flips or non-flips. The **Last Letter Concatenation** task (Wei et al., 2022b) requires the LLM to concatenate the last letters of each word in a given name list. Both tasks have out-of-domain test sets, which include more complex examples than those provided in the training exemplars. Specifically, for the Last Letter Concatenation task, the training set contains the lists comprising only two names but the test set consists of the lists with three and four names. A similar setting is adopted for the Coin Flip task.

**Tracking shuffled objects** (Srivastava et al., 2022) requires the LLM to determine the ultimate state of a system based on its initial condition and a series of subsequent changes. In each instance of this task, a collection of objects is initially associated with individual owners. These objects are then exchanged in a sequence of swappings. The model's objective is to determine the final owner of each object.

**Date Understanding** (Srivastava et al., 2022) asks the LLM to infer the date from the given context. The context is one or two sentences with date information.

**Navigate** (Srivastava et al., 2022) requires the LLM to infer the agent's position after several movements. Specifically, given a series of navigation steps, the LLM needs to determine whether or not an agent would end up back at the starting point.

**GSM8K** (Cobbe et al., 2021) is a collection of grade-level mathematics problems by human authors. These problems require 2 to 8 steps for resolution, primarily requiring a series of fundamental computations employing basic arithmetic operations (addition, subtraction, multiplication, and division) to arrive at the solution.

**Matrixshapes** (Srivastava et al., 2022) requires the LLM to predict the shape of the result of a chain of matrix manipulations, given the inputs' shapes.

**SVAMP** (Patel et al., 2021) is a challenge set for elementary-level Math Word Problems. It consists of questions that test a model across question sensitivity, reasoning ability, and invariance to structural alterations.

For each dataset, we sample three questions as the training set. Subsequently, we partition the remaining data into practice and test questions, maintaining a 1:4 ratio. Notice the CoinFlip and GSM8K are already split into the training set and the test set. We use the original test set and randomly sample 3 questions from the original training set to build our training set. Regarding Date Understanding, Navigate, Matrixshapes, and Tracking shuffled objects (5), we utilize data from (Cai et al., 2023). We divide each dataset into training, validation, and test sets, containing 3, 47, and 200 instances, respectively. For Auto-CoT, we select examples from the whole dataset in symbolic reasoning tasks and from the validation set in arithmetic reasoning tasks. Table 6 shows the example questions for each dataset.

## H  Experiment Setup

### H.1  Model Versions

In the experiments, we utilize two GPT models: gpt-3.5-turbo-16k and gpt-4. The gpt-3.5-16k refers to the "gpt-3.5-turbo-16k" model and in the OpenAI API model with checkpoint version 2023-06-13-preview webpage[1] GPT-4 refers to the "gpt-4" model with checkpoint version 2023-07-01-preview. All mentioned checkpoints are hosted on Microsoft Azure[2]. For Mixtral-8x7B-Instruct, we utilize "Mixtral-8x7B-Instruct-v0.1" on huggingface. The model temperature is 0 in all cases.

### H.2  Implementation Details

We use the default temperature of 0.0 for all models. For symbolic reasoning tasks, we utilize CoT as our base prompting method. For arithmetic tasks, we utilize the *Program of Thought* (PoT) method as our base prompting method. This method involves having the LLMs process natural language questions and create corresponding programs that represent intermediate reasoning steps, with the actual computation of solutions being delegated to a runtime environment, such as a Python interpreter. We avoid using CoT due to the tendency of language models to produce erroneous mathematical operation results when tackling arithmetic tasks (Ji et al., 2023). Our framework is designed to instruct the student model in problem-solving strategies rather than in performing detailed decimal operations with precision.

### H.3  Baselines

Since our framework generates a prompt for the student model for the downstream task, we compare several prompting methods in our experiments. Specifically, we adopt Zero-Shot CoT (Kojima et al., 2022), few-shot CoT (Wei et al., 2022b), and Auto-CoT (Zhang et al., 2022b) as baseline prompting methods. For the few-shot CoT, we use the questions in the training set as exemplars for few-shot prompting. For Auto-CoT, we use Sentence-BERT (Reimers & Gurevych, 2019) to compute a vector representation for each question and then cluster practice questions by *k*-means. Following the original paper, we choose the question closest to the center from each cluster and ask the teacher model to generate reasoning steps with zero-shot CoT to build final examples for the student model.

### H.4  Principle Lists

Table 7 and Table 8 show the principle list the teacher model finds in the error summarization stage for GPT3.5. These principles provide high-level guidance applicable across various

---

[1] .https://platform.openai.com/docs/models

[2] *.openai.azure.com

| **Coin Flip** |
|---|
| **Q**: A coin is heads up. Murraylee does not flip the coin. Meilich flips the coin. Is the coin still heads up? 
 **A**: no |

| **Last Letter Concatenation** |
|---|
| **Q**: Take the last letters of each word in "Maritza Nana Loretta Eric" and concatenate them. 
 **A**: "aaac" |

| **Tracking shuffled objects** |
|---|
| **Q**: Alice, Bob, Claire, Dave, and Eve are dancers at a square dance. At the start of a song, they each have a partner: Alice is dancing with Patrick, Bob is dancing with Sam, Claire is dancing with Jamie, Dave is dancing with Lola, and Eve is dancing with Melissa. Throughout the song, the dancers often trade partners. First, Dave and Eve switch partners. Then, Dave and Alice switch partners. Then, Eve and Alice switch partners. Then, Claire and Bob switch partners. Finally, Dave and Alice switch partners. At the end of the dance, Alice is dancing with 
 Options: 
 (A) Patrick 
 (B) Sam 
 (C) Jamie 
 (D) Lola 
 (E) Melissa 
 **A**: (A) |

| **Date Understanding** |
|---|
| **Q**: Jane scheduled 3 appointments with 5 people for tomorrow (Tue, 7/9/1972). What is the date a month ago in MM/DD/YYYY? 
 Options: 
 (A) 06/08/2059 
 (B) 06/22/1972 
 (C) 12/08/1971 
 (D) 06/08/2034 
 (E) 06/08/1972 
 (F) 06/07/1972 
 **A**: (E) |

| **Navigate** |
|---|
| **Q**: If you follow these instructions, do you return to the starting point? Always face forward. Take 1 step backward. Take 9 steps left. Take 2 steps backward. Take 6 steps forward. Take 4 steps forward. Take 4 steps backward. Take 3 steps right. 
 Options: 
 - Yes 
 - No 
 **A**: No |

| **GSM8K** |
|---|
| **Q**: Megan is an actress. She was the lead actress in 80% of her work. In total, Megan participated in 100 plays. How many times was Megan not the lead actress? 
 **A**: 20.0 |

| **SVAMP** |
|---|
| **Q**: In a school there are 308 girls and 318 boys. There are also 36 teachers. How many pupils are there in that school? 
 **A**: 626.0 |

| **Matrixships** |
|---|
| **Q**: Keep track of matrix shapes through various transformations. Transpose a matrix of shape (2,3,2). Transpose the result. Compute the Hadamard product of the result with a matrix of shape (2,3,2). Compute the Hadamard product of the result with a matrix of shape (2,3,2). Sum the result over the second axis. 
 **A**: (2,2) |

Table 6: Examples of questions in each dataset.

scenarios. However, the principle lists for GSM8K and SVAMP are less clear compared to others, since the errors in these tasks primarily relate to factual knowledge.

| Svamp |
| --- |
| 1. Incorrect Mathematical Operations: Ensure that the correct mathematical operations are used to solve the problem. Misinterpretation of the problem statement often leads to incorrect operations.
2. Misinterpretation of Problem Requirements: Understand the core requirement of the question.
3. Logical Errors in Variable Initialization: Be cautious when initializing variables. Make sure the initial values correctly represent the situation described in the question.
4. Misapplication of Variables in Calculation: Ensure that the variables are applied correctly in the calculation formula.
5. Understanding the Context of the Question: Contextual understanding is crucial. In the question about pots, flowers, and sticks, it's important to realize that the total number of flowers and sticks is a cumulative count across all pots, requiring multiplication of the per pot count by the total number of pots. |

| Date Understanding |
| --- |
| 1. Understanding Date Arithmetic and the 'datetime' Module: Several examples demonstrate a misunderstanding of how the 'datetime' module works, especially in terms of adding or subtracting days, months, and years. The 'timedelta' function in Python doesn't support months or years directly, so programmers need to account for this limitation when performing date arithmetic. Instead of using 'timedelta' for date manipulations, use 'dateutil.relativedelta'. This module provides more flexibility, especially for operations involving months and years.
2. Accurate Date Initialization: Initialize dates correctly. In several examples, the initial date is set without considering the context of the problem. Ensure that the starting point of the calculation aligns with the scenario's requirements.
3. Logical Consistency in Calculations: Maintain logical consistency in calculations. If the problem states a historical or future date, ensure that the calculations reflect this timeline accurately. Avoid mixing current dates ('datetime.now()') with historical or future scenarios unless it's relevant.
4. Validating Against Given Options: When comparing calculated dates against multiple-choice options, ensure that the options are correctly formatted and compared. It's essential to format the calculated date in the same format as the options for a valid comparison. |

| Navigate |
| --- |
| 1. Understand the Problem: Recognize that the problem requires tracking movements in two dimensions (horizontal and vertical). Understand that movements are influenced by the current direction the subject is facing. Identify the need to interpret turns as changes in direction, not just movement.
2. Initialize Variables: Define variables for horizontal and vertical movements, initializing them to zero. Introduce a variable for current direction, initializing it to the starting orientation (e.g., "north").
3. Extract and Interpret Instructions: Parse the given question to extract movement and turning instructions.
4. Use control structures (if-else, loops) to handle each instruction: For movement instructions (forward, backward), update the horizontal or vertical position based on the current direction. For turning instructions (right, left, around), update the current direction appropriately. 5. Handle Directional Changes: Implement logic to correctly modify the direction state when turning. For example, turning right from north means facing east.
6. Calculate Final Position: After processing all instructions, compare the final horizontal and vertical positions with the initial position (0,0).
7. Return Result: Return the appropriate option ("Yes" or "No") based on whether the final position matches the starting point.
8. Implement Directional Logic: Develop a mechanism to translate turning instructions into directional changes, affecting subsequent movement calculations.
9. Consider Special Cases: Account for any special or compound instructions that may require separate handling, ensuring all scenarios are covered. |

Table 7: Examples of principle list.

| GSM8K |
|---|
| 1. Incorrect Variable Assignment and Utilization: Ensure variable assignments accurately reflect the values they are supposed to represent based on the question's context.
2. Misinterpretation of Quantity Relationships: Accurately understand and interpret the relationships between different quantities as described in the problem statement.
3. Incorrect Mathematical Formulas: Ensure the mathematical formulas used align with the logical requirements of the problem.
4. Misinterpretation of Variable Values: Ensure variables are interpreted and utilized accurately based on the problem's context.
5. Omission of Critical Information: Incorporate all provided information and ensure no critical details are omitted in the solution.
6. Incorrect Arithmetic Operations: Ensure arithmetic operations are logically sound and mathematically correct. |
| **Matrixships** |
| 1. Matrix Multiplication: Use np.matmul for matrix multiplication.
2. Hadamard Product: Use * (asterisk) for element-wise multiplication (Hadamard product).
3. Transposition: Utilize np.transpose for matrix transposition. Do not specify the axes parameter when using np.transpose.
4. Resultant Matrix Shape: Always return the shape of the resulting matrix after an operation. |

Table 8: Examples of principle list.

### H.5 Problem-solving instruction examples

We provide an example of problem-solving instruction for a better understanding. The problem-solving instruction consists of a problem-solving method and several examples showing how to use the method. The questions in the examples are training questions.

### H.6 Overall student prompt examples

We provide an example of the overall teacher model's instruction for a better understanding. The overall instruction prompt consists of the modified problem-solving instruction and several newly selected examples from the validation set. Examples 1-3 are original examples in the problem-solving instruction (from the training set). Examples 4-6 are newly selected examples from the validation set.

Listing 1: An example of problem-solving instructions provided by the teacher model.

```
1  ###Problem-solving instruction for CoinFlip task
2
3  Method to solve coin filp problems:
4  1. Start by reading the initial state of the coin (heads up or tails up).
5  2. Examine the actions of each person mentioned in the question in the
        order they are mentioned. If a person flips the coin, the state of
        the coin will change. If it was heads up, it will now be tails up,
        and vice versa.
6  If a person does NOT flip the coin, the state remains unchanged. At the
        end of the actions, state the final position of the coin as 'heads up
        ' or 'tails up'.
7  3. Answer the question based on the final state of the coin compared to
        the state asked in the question.
8
9  Examples:
10
11 Question: A coin is heads up. sager does not flip the coin. zyheir flips
        the coin. Is the coin still heads up?
12 Explanation:
13 The coin starts as heads up.
14 Sager does not flip the coin, so it remains heads up.
15 Zyheir flips the coin, changing its state. Now, it's tails up.
16 The final position of the coin is tails up.
17
18 Answer: No.
19
20 Question: A coin is heads up. mailey does not flip the coin. maurisa does
         not flip the coin. Is the coin still heads up?
21 Explanation:
22 The coin starts as heads up.
23 Mailey does not flip the coin, so it remains heads up.
24 Maurisa also does not flip the coin, so it stays heads up.
25 The final position of the coin is heads up.
26
27 Answer: Yes.
28
29 Question: A coin is heads up. murraylee does not flip the coin. meilich
        flips the coin. Is the coin still heads up?
30
31 Explanation:
32 The coin starts as heads up.
33 Murraylee does not flip the coin, so it remains heads up.
34 Meilich flips the coin, changing its state. Now, it's tails up.
35 The final position of the coin is tails up.
36 Answer: No.
```

Listing 2: An examples of overall prompt provided by the teacher model.

```
1  ###Overall prompts for 'Navigate' task
2
3  To write Python functions to answer questions about returning to the
        starting point after following a set of movement instructions, follow
         these steps:
4
5  Understand the Problem:
6  Recognize that the problem requires tracking movements in two dimensions
        (horizontal and vertical).
7  Understand that movements are influenced by the current direction the
        subject is facing.
8  Identify the need to interpret turns as changes in direction, not just
        movement.
9
10 Design the Function:
11
```

```
12      Initialize Variables:
13      Define variables for horizontal and vertical movements, initializing
         them to zero.
14      Introduce a variable for current direction, initializing it to the
         starting orientation (e.g., "north").
15      Extract and Interpret Instructions:
16      Parse the given question to extract movement and turning instructions
         .
17      Use control structures (if-else, loops) to handle each instruction:
18      For movement instructions (forward, backward), update the horizontal
         or vertical position based on the current direction.
19      For turning instructions (right, left, around), update the current
         direction appropriately.
20      Handle Directional Changes:
21      Implement logic to correctly modify the direction state when turning.
          For example, turning right from north means facing east.
22      Calculate Final Position:
23      After processing all instructions, compare the final horizontal and
         vertical positions with the initial position (0,0).
24      Return Result:
25      Return the appropriate option ("Yes" or "No") based on whether the
         final position matches the starting point.
26
27  Implement Directional Logic:
28  Develop a mechanism to translate turning instructions into directional
         changes, affecting subsequent movement calculations.
29
30  Test the Function:
31  Write test cases to execute the function with different sets of
         instructions.
32  Store and verify the result in a variable answer to ensure the function
         behaves as expected.
33
34  Here are some examples:
35
36  Question: 'If you follow these instructions, do you return to the
         starting point? Always face forward. Take 1 step backward. Take 9
         steps left. Take 2 steps backward. Take 6 steps forward. Take 4 steps
          forward. Take 4 steps backward. Take 3 steps right. Options: - Yes -
          No'
37
38  Answer:
39  ```python
40  def do_you_return_to_starting_point():
41      # Initial position and direction
42      horizontal, vertical = 0, 0
43      direction = "north"
44
45      # Movement instructions
46      instructions = [
47          ("backward", 1),
48          ("left", 9),
49          ("backward", 2),
50          ("forward", 6),
51          ("forward", 4),
52          ("backward", 4),
53          ("right", 3)
54      ]
55
56      # Process each instruction
57      for action, steps in instructions:
58          if action == "forward":
59              if direction == "north":
60                  vertical += steps
61          elif action == "backward":
```

```python
 62              if direction == "north":
 63                  vertical -= steps
 64          elif action == "left":
 65              if direction == "north":
 66                  horizontal -= steps
 67          elif action == "right":
 68              if direction == "north":
 69                  horizontal += steps
 70
 71      # Check if returned to the starting point
 72      return "Yes" if horizontal == 0 and vertical == 0 else "No"
 73
 74 # Execute the function
 75 answer = do_you_return_to_starting_point()
 76 ```
 77 By using the python function above, you can get the answer of the
       question.
 78 ###END
 79
 80 Question: 'If you follow these instructions, do you return to the
       starting point? Always face forward. Take 10 steps left. Take 10
       steps forward. Take 7 steps forward. Take 2 steps forward. Options: -
        Yes - No'
 81
 82 Answer:
 83 ```python
 84 def do_you_return_to_starting_point():
 85      # Initial position and direction
 86      horizontal, vertical = 0, 0
 87      direction = "north"
 88
 89      # Movement instructions
 90      instructions = [
 91          ("left", 10),
 92          ("forward", 10),
 93          ("forward", 7),
 94          ("forward", 2)
 95      ]
 96
 97      # Process each instruction
 98      for action, steps in instructions:
 99          if action == "forward":
100              if direction == "north":
101                  vertical += steps
102          elif action == "left":
103              if direction == "north":
104                  horizontal -= steps
105
106      # Check if returned to the starting point
107      return "Yes" if horizontal == 0 and vertical == 0 else "No"
108
109 # Execute the function
110 answer = do_you_return_to_starting_point()
111 ```
112 By using the python function above, you can get the answer of the
       question.
113 ###END
114
115 Question: 'If you follow these instructions, do you return to the
       starting point? Always face forward. Take 1 step right. Take 3 steps
       left. Take 2 steps right. Options: - Yes - No'
116
117 Answer:
118 ```python
119 def do_you_return_to_starting_point():
```

```
120     # Initial position and direction
121     horizontal, vertical = 0, 0
122     direction = "north"
123
124     # Movement instructions
125     instructions = [
126         ("right", 1),
127         ("left", 3),
128         ("right", 2)
129     ]
130
131     # Process each instruction
132     for action, steps in instructions:
133         if action == "left":
134             if direction == "north":
135                 horizontal -= steps
136         elif action == "right":
137             if direction == "north":
138                 horizontal += steps
139
140     # Check if returned to the starting point
141     return "Yes" if horizontal == 0 and vertical == 0 else "No"
142
143 # Execute the function
144 answer = do_you_return_to_starting_point()
145 ```
146 By using the python function above, you can get the answer of the
        question.
147 ###END
148
149 Question: 'If you follow these instructions, do you return to the
        starting point? Take 3 steps. Turn around. Take 3 steps. Turn right.\
        nOptions:\n- Yes\n- No'
150
151 Answer:
152 ```python
153 def do_you_return_to_starting_point():
154     # Initial position and direction
155     horizontal, vertical = 0, 0
156     direction = "north"
157
158     # Movement instructions
159     instructions = [
160         ("forward", 3),
161         ("turn", "around"),
162         ("forward", 3),
163         ("turn", "right")
164     ]
165
166     # Process each instruction
167     for action, value in instructions:
168         if action == "forward":
169             if direction == "north":
170                 vertical += value
171             elif direction == "south":
172                 vertical -= value
173             elif direction == "east":
174                 horizontal += value
175             elif direction == "west":
176                 horizontal -= value
177         elif action == "turn":
178             if value == "around":
179                 direction = "south" if direction == "north" else "north"
180             elif value == "right":
181                 if direction == "north":
```

```
182                    direction = "east"
183                elif direction == "east":
184                    direction = "south"
185                elif direction == "south":
186                    direction = "west"
187                elif direction == "west":
188                    direction = "north"
189
190      # Check if returned to the starting point
191      return "Yes" if horizontal == 0 and vertical == 0 else "No"
192
193  # Execute the function
194  answer = do_you_return_to_starting_point()
195  ```
196  By using the python function above, you can get the answer of the
         question.
197  ###END
198
199  Question: 'If you follow these instructions, do you return to the
         starting point? Take 3 steps. Turn around. Take 5 steps. Turn right.
         Turn right. Take 1 step. Take 1 step.\nOptions:\n- Yes\n- No'
200
201  Answer:
202  ```python
203  def do_you_return_to_starting_point():
204      # Initial position and direction
205      horizontal, vertical = 0, 0
206      direction = "north"
207
208      # Movement instructions
209      instructions = [
210          ("forward", 3),
211          ("turn", "around"),
212          ("forward", 5),
213          ("turn", "right"),
214          ("turn", "right"),
215          ("forward", 1),
216          ("forward", 1)
217      ]
218
219      # Process each instruction
220      for action, value in instructions:
221          if action == "forward":
222              if direction == "north":
223                  vertical += value
224              elif direction == "south":
225                  vertical -= value
226              elif direction == "east":
227                  horizontal += value
228              elif direction == "west":
229                  horizontal -= value
230          elif action == "turn":
231              if value == "around":
232                  direction = "south" if direction == "north" else "north"
233              elif value == "right":
234                  if direction == "north":
235                      direction = "east"
236                  elif direction == "east":
237                      direction = "south"
238                  elif direction == "south":
239                      direction = "west"
240                  elif direction == "west":
241                      direction = "north"
242
243      # Check if returned to the starting point
```

```
244      return "Yes" if horizontal == 0 and vertical == 0 else "No"
245
246  # Execute the function
247  answer = do_you_return_to_starting_point()
248  ```
249  By using the python function above, you can get the answer of the
         question.
250  ###END
251
252  Question: 'If you follow these instructions, do you return to the
         starting point? Take 1 step. Take 5 steps. Turn around. Turn around.
         Turn around. Take 6 steps.\nOptions:\n- Yes\n- No'
253
254  Answer:
255  ```python
256  def do_you_return_to_starting_point():
257      # Initial position and direction
258      horizontal, vertical = 0, 0
259      direction = "north"
260
261      # Movement instructions
262      instructions = [
263          ("forward", 1),
264          ("forward", 5),
265          ("turn", "around"),
266          ("turn", "around"),
267          ("turn", "around"),
268          ("forward", 6)
269      ]
270
271      # Process each instruction
272      for action, value in instructions:
273          if action == "forward":
274              if direction == "north":
275                  vertical += value
276              elif direction == "south":
277                  vertical -= value
278              elif direction == "east":
279                  horizontal += value
280              elif direction == "west":
281                  horizontal -= value
282          elif action == "turn":
283              if value == "around":
284                  direction = "south" if direction == "north" else "north"
285
286      # Check if returned to the starting point
287      return "Yes" if horizontal == 0 and vertical == 0 else "No"
288
289  # Execute the function
290  answer = do_you_return_to_starting_point()
291  ```
292  By using the python function above, you can get the answer of the
         question.
293  ###END
```

