# OpenReview forum: "TPD: Enhancing Student Language Model Reasoning via Principle Discovery and Guidance"
_colmweb.org/COLM/2024/Conference — COLM_

### Official Review · Reviewer_ZvuT · 2024-05-08

**Rating:** 4
**Confidence:** 5
**Ethics Flag:** 1

**Summary:**

This paper proposes a simple method to improve the performance of the student model. Specifically, the teacher model generates problem-solving instructions and corrective principles based on the student LLM's errors. Then during inference, the student model can guide the inference process without the help from the teacher model.

**Questions To Authors:**

Please refer to the reasons to reject.

**Reasons To Accept:**

1. The proposed method is intuitive which imitates the real teaching scenarios in schools.
2. The analysis of the experiments is detailed and convincing.

**Reasons To Reject:**

1. The method involves the labeling process of both large teacher model and human reviewers, which is complex and resource-consuming.
2. The experimental result on some arithmetic reasoning tasks achieves a very subtle improvement. More seriously, it seems that some important baselines are missing, such as Automate-CoT.
3. The experiments are only conducted on GPT-3.5, which is a relatively weak LLM. It would be better if the authors could provide the experimental results on other LLMs.

---

> ### Author Rebuttal · Authors · 2024-05-31
>
> **1. The method involves the labeling process of both large teacher model and human reviewers, which is complex and resource-consuming.**
>
> Our work primarily explores the interactions between LLMs, aiming to study how an advanced LLM can effectively teach a weaker LLM through interaction. The involvement of a large teacher model is essential for this process, as it aligns with the standard practice in other studies [1][2] focusing on the teacher-student framework.
>
> While it is true that human reviewers are necessary to ensure the quality of the principles summarized by the teacher model, their role is limited to identifying and removing any erroneous or confusing principles. This task, while important, is simple and manageable. The human reviewers' involvement ensures the reliability and accuracy of the learning process, ultimately enhancing the effectiveness of our approach.
>
> **2. The experimental result and Baseline**
>
> We appreciate the reviewer’s feedback. TPD can guide student models on how to solve these types of questions through principles but cannot provide factual knowledge for each specific question due to the use of a fixed prompt during inference time. The performance gain on some arithmetic reasoning tasks is subtle, primarily because of this lack of factual knowledge. This limitation can be addressed by incorporating a retrieval module during the student model's inference stage to retrieve prior experiences and feedback from practice questions. We plan to include related experiments in the appendix of the next version.
>
> Regarding the baselines, Automate-CoT is capable of selecting the optimal combination of several rationale chains, which can also be implemented within a teacher-student framework. We will include Automate-CoT as a new baseline in the next version.
>
> **3. The experiments are only conducted on GPT-3.5**
>
> We would like to kindly point out that we have also conducted experiments on Mixtral 8x7B. The results are shown in Table 1. We also plan to add llama3 models as students in the revised version.
>
>
> [1] Ho, Namgyu, Laura Schmid, and Se-Young Yun. "Large language models are reasoning teachers." arXiv preprint arXiv:2212.10071 (2022).
>
> [2] Saha, Swarnadeep, Peter Hase, and Mohit Bansal. "Can language models teach weaker agents? teacher explanations improve students via theory of mind." arXiv preprint arXiv:2306.09299 (2023).

---

> > ### Comment · Reviewer_ZvuT · 2024-06-06
> >
> > Thanks for your response. It's better to add Llama3 models as students in the next version.

---

> > > ### Author Response · Authors · 2024-06-07
> > >
> > > Dear Reviewer ZvuT,
> > >
> > > We greatly appreciate the time you took to review our paper. Since the Llama3 models are released on April 18th, which is later than the paper submission date, we will add them in the next version.
> > >
> > > Due to the short duration of the author-reviewer discussion phase, we would appreciate your feedback on whether your main concerns have been adequately addressed. Should you have any further advice on the paper and/or our rebuttal, please let us know and we will be more than happy to engage in more discussions.
> > >
> > > Thank you so much for devoting time to improving our paper!

---

### Official Review · Reviewer_AV4e · 2024-05-11

**Rating:** 6
**Confidence:** 4
**Ethics Flag:** 1

**Summary:**

Large language models (LLMs) possess substantial reasoning abilities; however, they entail significant computational demands. Prior research has explored knowledge-distillation-based fine-tuning methods to confer these capabilities onto smaller models, though these methods are resource-intensive. This paper presents a novel approach, Teaching via Principle Discovery (TPD), which enhances the reasoning skills of smaller language models (student LLMs) by leveraging insights from larger models (teacher LLMs) without necessitating ongoing intervention during inference. TPD adopts human teaching methodologies through a two-phase procedure: principle generation and principle exploitation, guided by error analysis and the formulation of corrective principles by the teacher model.

**Reasons To Accept:**

1. The introduced framework is innovative, proposing principle discovery as an interesting new direction for transferring the reasoning powers of LLMs to smaller models.

2. The method is not just innovative, but also rigorously designed. The two-phase process of principle generation and exploitation is meticulously planned, effectively emulating human instructional techniques. This careful design ensures the method's reliability and validity.

3. The empirical evidence demonstrates notable enhancements over conventional methods such as chain-of-thought prompting, making a strong case for TPD's effectiveness.

**Reasons To Reject:**

1. Inclusion of additional case studies would enhance the paper. While TPD demonstrates promising results for symbolic and arithmetic reasoning tasks, a more detailed exposition with extensive examples could clarify its advantages to readers.

2. The paper should introduce and discuss failure scenarios more. Currently, the limitations of TPD and its applicability to tasks beyond the tested domains are not sufficiently discussed.

3. Discussions on how well the principles learned can be generalized and transferred to different student models or a wider array of tasks are limited and could be expanded.

---

> ### Author Rebuttal · Authors · 2024-05-31
>
> **1. Inclusion of additional case studies would enhance the paper.**
>
> Thanks for pointing out. We will add a new figure to visualize and compare the original examples, extensive examples, and the principle list for a better understanding in the next version.
>
> **2. The paper should introduce and discuss failure scenarios more.**
>
> We appreciate the reviewer’s feedback and acknowledge the importance of discussing the failure scenarios and limitations of TPD.
>
> Firstly, TPD effectively teaches student models how to approach problem-solving through principles. However, it does have a limitation in imparting factual knowledge for specific questions due to the fixed prompt used during inference. This limitation can be mitigated by incorporating a retrieval module in the student model's inference stage. This module would retrieve prior experiences and feedback from practice questions, enhancing the model's ability to provide factual answers.
>
> Secondly, we believe that TPD has the potential to be applied to other domains, such as complex reasoning tasks and agent-based tasks. Principles serve as intuitive guides for human thinking and planning, and LLMs can similarly benefit from these principles to improve their performance in diverse and complex domains.
>
> **3. Discussions on how well the principles learned can be generalized and transferred**
>
> We will compare different baselines and extend Table 3 in the next version. Meanwhile, we will also visualize a comparison between principles for two LLMs for a better understanding of the transferable setting.

---

### Official Review · Reviewer_Yx4q · 2024-05-12

**Rating:** 6
**Confidence:** 4
**Ethics Flag:** 1

**Summary:**

The paper proposes a method to design task-specific zero-shot or few-shot prompts through interaction between a "teacher" and "student" model, using an existing dataset of task-specific question-answer pairs. This involves a multi-step process summarized below:
1. The teacher model selects representative samples from the dataset and generates an initial instruction on how to solve the task, along with few-shot examples of how to solve the selected samples. (this is the baseline prompt denoted TPD w/o ES in Table 1, to my understanding)
2. The student model is prompted to solve questions in a validation set using the initial prompt constructed above, and an error set $E$ is built.
3. The teacher model is prompted to solve the same questions in $E$. Each question that the teacher can solve correctly is included in the feasible error set $\hat{E}$. Incorrect questions are discarded as they are deemed too difficult for even the teacher.
4. The teacher model build a set of principles $P$ that is required to address the errors in $\hat{E}$ through an iterative process.
5. The resulting principle list is filtered by human reviewers.
6. A new set of representative questions are selected from the validation set, which violate the most number of principles in $P$.
7. The initial instruction from step 1 is revised based on $P$, and new examples of how to solve the questions selected in step 6 are appended to the original examples from step 1 (this is the prompt denoted TPD w/ ES in Table 1, to my understanding).

The final prompt outperforms zero-shot, few-shot, and auto-cot on various symbolic and arithmetic reasoning tasks.

**Questions To Authors:**

1. Is there a specific reason behind the selection of Mixtral 8x7B–an MoE model–as the sole open-source student model?

**Reasons To Accept:**

1. The proposed method outperforms previous prompt construction methods on a wide range of reasoning tasks across two different student models.

**Reasons To Reject:**

1. The requirement of large task-specific datasets, and thus the associated costs, limits the applicability of this method. It may be more cost-effective to fine-tune a smaller model to achieve the same performance. This has not been considered in the paper. Note, while Auto-CoT has the same limitations, they mitigate this limitation by demonstrating a streaming-based variant of their approach, where examples are selected from test batches.
1. The need for human intervention in step 5 of my summary somewhat limits the applicability of this method.
1. The difference between the experimental settings of Figure 3 (a), (b) vs (c) is unclear. Both seem to ablate the number of examples.
1. It is hard to conclude that models *benefit* from principles deduced from errors of another model from Table 3. That would require comparison with baselines (e.g., zero-shot CoT, maybe few-shot CoT, principles derived from original samples, etc.).
1. (Minor) The method is fairly complicated.
1. (Minor) The writing is hard to follow. E.g.,
    1. The method explanation and Figure 2 are hard to follow. It may be beneficial to divide the method into a clear series of steps which align with the subsections of the method section.
    1. It is not 100% clear what TPD w/ and w/o ES in Table 1 refers to (I've written my interpretation in the Summary)
    1. It is hard to follow which parts of the methodology explained in Section 3 corresponds to which ablations in Table 2, and it is unclear what method was used in Figure 3 and Table 3.
    1. The use of the word "teaching" throughout the abstract and intro may mislead the reader to initially believe this is a student fine-tuning paper. It would help the reader to highlight that the method's goal is to construct a (few-shot) prompt.

---

> ### Author Rebuttal · Authors · 2024-05-31
>
> **1. The requirement of large task-specific datasets,...**:
>
> In our settings, we do not consider finetuning baselines as they are time-consuming. In this study, we mainly explore the interactions between language models. We want to study how to let an advanced LLM teach a weaker LLM through interaction effectively. We discussed finetuning methods and their limitations in the related works. In TPD, the additional computational cost mainly comes from the teacher model.  TPD can also work streaming setting where a small batch of test questions (say m questions) arrive at a time like in data streams.
>
> Initialize $M_0$; When batch 1 of questions $q_1^1$ ,...,$q_1^m$ arrive, the teacher model provides a problem-solving instruction based on these questions and the student model can utilize the instruction for each $q_1^i$ to obtain its answer $a_1^i$. Add these question-answer pairs $[(q_1^1, a_1^1), (q_1^2, a_1^2), (q_1^m, a_1^m)]$ to $M_0$ and call the new set $M_1$.
>
> When batch $b$ (b>1) of questions $q_b^1 ,...,q_b^m$ arrive, the teacher model can build a principle list $P$ based on $M_{b-1}$ as described in TPD, and then utilize $P$ to select examples and modify the problem-solving instructions to build a final prompt for this batch of questions. The student model then uses the demonstrations for each question $q_b^i$ to get an answer $a_b^i$. Finally, add these question-answer pairs $[(q_b^1, a_b^1), (q_b^2, a_b^2), (q_b^m, a_b^m)]$ to $M_{b-1}$ and call the new set $M_b$.
> We’ll add these details to the revised manuscript.
>
> **2.The need for human intervention**
>
> Human intervention is necessary because the teacher model may not always accurately summarize the meaningful principles derived from the student model's mistakes. Human reviewers are involved only in deleting any principles that are erroneous or confusing. This task is straightforward and does not impose a significant burden, thus maintaining the method's overall efficiency.

---

> > ### Author Response · Authors · 2024-05-31
> >
> > **3. Figure 3 (a), (b) vs (c)**
> >
> > For Figure 3(a) and (b), we study whether we need to have examples in the problem-solving instruction in the principle generation stage. If we do not have examples in the problem-solving instruction, it will only contain a problem-solving method. The results show that examples are very important in problem-solving instruction.
> > For Figure 3(c), we study the number of examples selected in the principle exploitation stage, the examples here are selected from the practice questions. We want to study the minimum number of examples required for effective learning from the principle list.
> >
> > **4. It is hard to conclude that models benefit from principles deduced from errors of another model from Table 3.**
> >
> > We will compare different baselines and extend Table 3 in the next version. Meanwhile, we will also visualize a comparison between principles for two LLMs for a better understanding.
> >
> > **5. (Minor) The method is fairly complicated.**
> >
> > Our method is designed to mimic natural human teaching and learning processes. The goal is to explore how an advanced LLM can effectively teach a weaker LLM through interaction. While the additional computational cost primarily arises from the teacher model, we anticipate that as the costs of advanced models continue to decrease, the cost of our TPD approach will become more manageable.
> >
> >
> > **6. The writing is hard to follow.**
> >
> > Thanks for pointing out. We will double-check and revise the writing.
> >
> > **7. Is there a specific reason behind the selection of Mixtral 8x7B–an MoE model–as the sole open-source student model?**
> >
> > No, we just select it and use it. We will also add llama3-70B as a new student model in the revised version.

---

> > ### Comment · Reviewer_Yx4q · 2024-05-31
> >
> > Thank you for acknowledging my comments.
> >
> > - 1. I'd like to point out that the requirement of an existing large dataset is a large cost factor, which may be overlooked when applying to academic benchmarks--since they implicitly provide a large dataset with them. In novel tasks, practitioners will have to build these datasets prior to applying the proposed method. Therefore the dataset should be considered as a cost associated to this method.
> >
> > - 3. I see that the explanation for Figure 3 (a) and (b) in the paper ( `numbers of examples in the problem-solving instruction`) is clear but (c) could be clarified better. I also think that establishing the names of each method/step more clearly (mentioned in Reasons to Reject 6.1.) can prevent this misunderstanding.
> >
> > I will maintain my initial rating.

---

### Official Review · Reviewer_R4tU · 2024-05-16

**Rating:** 6
**Confidence:** 4
**Ethics Flag:** 1

**Summary:**

This paper proposes a method to effectively transfer advanced reasoning capabilities of stronger LLMs to weaker LLMs by exploiting general principles that indicate gaps in a student’s understanding on practice problems. They outperform CoT prompting on symbolic and arithmetic reasoning.

**Questions To Authors:**

Please go through the reasons to reject, and you can respond. If I receive clarifying satisfactory answers then I am willing to increase my rating.

**Reasons To Accept:**

- The idea of principle discovery and exploitation is novel (though there are other recent related work (e.g., cite: LEAP [1]), but we can discount those given they are just 3 months old though eventually they must be included).
- The setting is nicely grounded conceptually in human teaching.
- The method does not require the expensive teacher model at inference time.
- The paper is generally well-written.

**Reasons To Reject:**

**Availability and selection of training examples**:
The paper assumes that the training examples are available in the principle generation stage (constituting a validation set for initial practice) . A realistic setting for real-world tasks and benchmarks like Big Bench Hard is where typically only a handful of examples are available. How many examples are sufficient for helpful problem solving instructions and error summarization? Is there a method to optimally select from the training examples e.g. with a higher coverage of question types (or principle types)?

What about other tasks like open ended generation. In many such real world examples, it is not easy to compare against the ground truth and measure most error prone questions. In those practical setups, how will you choose examples based on the principle list containing the most error-prone questions for the student model to help the student model learn from errors effectively (in section 4.4 last para, it is stated that most error prone questions are required for effective learning in the principle generation stage)?


**Issues in Principle exploitation**:
Generated Principles in Figure 4 are too high level ("_Logic flow: ensure arithmetic operations are logically sound and mathematically correct_"). Is this example really providing any actionable insights to a model during inference? How do you know which kind of error is triggered so that you can select a principle? Also fundamentally, how are principles different from feedback?  Other similar papers (e.g., cite: CLIN [5]) that have a two stage pipeline that includes “error summarization” that they call meta memory. But their summarized reflections are distilled into knowledge necessary to solve the task.

Related to this: How do you pull past principles (did you try different retrieval methods)? Past similar methods (cite: Memprompt) that exploit past mistakes for better inference have documented the importance of retrieval quality. How will you scale the method as the principle list grows?

**Missing error analysis**:
It would be helpful to know the kind of errors the system makes. e.g., what percentage of the errors are related to principle generation and exploitation. There are many details related to principle retrieval issues, teacher making mistakes in error identification or just the fact that a principle is unhelpful or the model fails despite providing a helpful principle. The ablation studies were helpful (thank you), but error analysis would give a clear picture.

**Missing Related work and Baselines**:
There has been past work on exploiting model answers to create a memory/ list of knowledge (cite: CLIN [5]), feedback (cite: MemPrompt [4]), principles (cite: LEAP [1]). Also, Reflexion [2] learns per-example feedback and LEAP [1] learns per-task principles. Can you compare against these methods, either empirically or theoretically? It seems like some of this is very closely related work (e.g., LEAP [1]) and it would be good to compare with these (though empirical comparison with LEAP [1] is not at all a ground for rejection as it is a very new paper but it would be useful to compare it broadly).

Section 4.4 page 8 presents an interesting finding that the LLM diverges significantly from its previous answer if a critique or feedback is used. Is this a previously established finding / can you support this claim with any references? This looks contrary to other papers (e.g., cite: [3] page 7) so it is important to substantiate.

Also, please distinguish your work in the related work, in addition to mentioning other papers. Eg., it seems like you also search for principles applied over mistakes from most representative train examples? (so there must be a retrieval module here like other papers?)

**Results are not very convincing**:
The claim made in the paper are quite broad. However, the empirical results are not too convincing. The improvement over AutoCoT or 3-shot CoT is not significant and it is raises the question whether we can justify a more complex method like yours. It would be helpful to also have a simple baseline that takes the k-examples as demonstrations whose principles are applied in your method.


**Other Missing details**:
On page 4, the teacher can make mistakes in accurately identifying the incorrect answers in the error set. Can you clarify if the teacher sometimes cannot give the correct modified instructions or the correct principle? Can the teacher identify when an answer is wrong (the paper mentions that these examples are taken from a validation set (question, answer) -- so why is it hard to detect if there is an error e.g., in math, the answers are clear (but maybe in open ended generation it is not easy to identify mistakes). I am assuming that at test time, there is no teacher involved as mentioned in the paper.


**Minor typos**:
typo in fig 1: promblem (I have not checked more thoroughly for spelling mistakes elsewhere)


References:
1. In-Context Principle Learning from Mistakes (LEAP): https://arxiv.org/pdf/2402.05403
2. Reflexion: Language agents with verbal reinforcement learning: https://arxiv.org/abs/2303.11366
3. Learning to Repair: Repairing model output errors after deployment using a dynamic memory of feedback: https://arxiv.org/pdf/2112.09737
4. Memory-assisted prompt editing to improve GPT-3 after deployment: https://arxiv.org/abs/2201.06009
5. CLIN: A Continually Learning Language Agent for Rapid Task Adaptation and Generalization: https://arxiv.org/abs/2310.10134

---

> ### Author Rebuttal · Authors · 2024-05-31
>
> **1. Availability and selection of training examples**:
>
> We appreciate your pointing this out and agree with your judgment. In our experimental settings, we randomly select 25% of the data from the training set to serve as the validation set. A larger set of practice questions enhances principle generation. The table below shows the influence of the size of the validation set on TPD’s performance (on GPT-3.5).
>
> |                | 5%  | 15% | 25% | 35% |
> |----------------|-----|-----|-----|-----|
> | **Navigation** | 86.5| 92.5| 97.5| 97.5|
> | **Matrix**     | 88.0| 91.5| 93.5| 94.0|
>
> We observe that increasing the size of the validation set indeed improves the framework's performance. If we had only 3 examples in the validation set, the framework would degenerate into few-shot learning.
>
> We believe that a higher coverage of principle types can help optimally select training examples. To demonstrate this, we compare the results from different numbers of principles:
>
> |                | Half Principle List | Full Principle List |
> |----------------|---------------------|---------------------|
> | **Navigation** | 93.0                | 97.5                |
> | **Matrix**     | 89.5                | 93.5                |
>
> Our TPD method selects examples based on the number of violations against the principle list. However, we can modify it to ensure higher coverage of the principle list, further optimizing example selection.
>
> For open-ended generation tasks, there is no single correct answer for each question. However, there are principles related to the quality or human preference of the generation, such as factuality and helpfulness. In these settings, we can choose examples based on questions that are prone to violating these principles. These scenarios are more akin to alignment tasks, which cannot be easily addressed solely through prompting.
>
> **We really appreciate the reviewer's valuable feedback. Since there are many questions to be discussed, we put the remaining response in the following anonymous link**
>
> https://docs.google.com/document/d/1MOgF8-LhPx6fd5knME4favjqYEJ0XqrzAJ_aauj0hEg/edit?usp=sharing
>
> **We do not add a new revision or a large amount of new content in this anonymous file. It only contains responses to the reviewer's questions.**

---

> > ### Author Response · Authors · 2024-05-31
> >
> > **Q2. Issues in Principle exploitation**
> >
> > We would like to clarify the following points:
> >
> > Principle Utilization: Our approach does not directly put the principles into the prompt. Instead, we select informative examples based on these principles, allowing the student model to learn logical coherence from these examples. TPD does not select different examples for different questions during the student model's inference. It utilizes a fixed principle list discovered during student practice to select examples from the validation set to build a fixed prompt for all instances from the same task.
> >
> > Experiment Method: In our experiments, we put the selected examples into the teacher-generated instructions for all inference questions. Our goal is to build a teacher-student framework for offline settings, where the student can generally utilize the principles taught by the teacher without other help.
> >
> > Principles vs. Feedback: Principles provide high-level abstract guidance, teaching the student model methods to solve one type of problem. In contrast, feedback methods offer detailed guidance for each specific question, which needs to be retrieved during the student model’s inference. TPD tests how high-level principles can help the student model and it can also be augmented with previous feedback. We will add the retrieval module to TPD and show the results in the next version.
> >
> > **Q3. Related to this: How do you pull past principles? How will you scale the method as the principle list grows?**
> >
> > In our experiments, we treat principles as high-level guidance from the teacher model. The teacher model constructs a fixed prompt in TPD for all instances in a specific type of task, and we evaluate how effectively the student model learns from this prompt.
> >
> > Retrieval methods, on the other hand, retrieve different feedback from past experiences, offering more fine-grained and question-specific guidance. To draw an analogy, principles are akin to what teachers impart during class, while feedback represents the past experiences encountered by the student model.
> >
> > In Memprompt, the memory of past experiences expands during the inference stages. However, in TPD, we work with a fixed set of principles derived from practice questions. Although TPD currently uses a static list of principles, it can be augmented with feedback and retrieval methods. We plan to explore this enhancement in future iterations of our work.
> >
> > **Q4. Missing error analysis: It would be helpful to know the kind of errors the system makes.**
> >
> > We have conducted an error analysis, and the results are summarized in the following tables.
> > Number of Failure cases:
> >
> > |         | Coin | Letter | Shuffled | Date | Navi. | GSM8K | Matrix | SVAMP |
> > |---------|------|--------|----------|------|-------|-------|--------|-------|
> > | GPT-3.5 | 0    | 0      | 1        | 2    | 1     | 3     | 1      | 2     |
> > | Mixtral | 0    | 0      | 0        | 0    | 1     | 4     | 2      | 2     |
> >
> > Number of Principles That Do Not Improve Performance:
> > |         | Coin | Letter | Shuffled | Date | Navi. | GSM8K | Matrix | SVAMP |
> > |---------|------|--------|----------|------|-------|-------|--------|-------|
> > | GPT-3.5 | 0    | 0      | 0        | 1    | 0     | 2     | 0      | 2     |
> > | Mixtral | 0    | 0      | 0        | 0    | 0     | 1     | 0      | 2     |
> >
> > The first table illustrates the number of failure cases identified. The teacher model is effective in identifying the student's failures, showing a relatively low number of failures. This highlights the necessity of using the filtering module in TPD to filter out these cases.
> >
> > The second table displays the number of principles that do not improve the student model's performance. Some principles do not aid in arithmetic reasoning tasks, possibly because they are too general.
> >
> > Due to the length constraints, we will include a detailed error analysis section, including a comprehensive breakdown of the types of errors and their causes, to offer a clearer picture in the revised paper.

---

> > > ### Author Response · Authors · 2024-05-31
> > >
> > > **Q5. Missing Related work and Baselines**
> > >
> > > We appreciate your highlighting the importance of comparing our work with previous studies. We acknowledge that there has been significant past work on leveraging model answers to create a memory or list of knowledge, as well as feedback mechanisms and principles. Specifically, CLIN [5] focuses on retrieving successful trials from history, MemPrompt [4] retrieves the most similar questions with user feedback in the history, and Reflexion [2] uses summaries of previous failures. LEAP [1], the most closely related work, summarizes high-level and low-level principles from the model’s previous mistakes.
> > >
> > > Comparison with LEAP [1].
> > > LEAP [1] and our approach share similarities in that both aim to discover principles from previous mistakes. However, there are two main differences between our methods:
> > >
> > > 1. Framework Application: We apply principle discovery within a teacher-student framework to enhance knowledge transfer through principles. In contrast, LEAP demonstrates that LLMs can also discover principles from their previous mistakes.
> > >
> > > 2. Application of Principles: We apply the principles through new examples from the validation set, whereas LEAP directly concatenates principles to the original prompt.
> > >
> > > Comparison with Reflexion [2], CLIN [5], and MemPrompt [4].
> > > These methods utilize per-example feedback. Specifically:
> > >
> > > 1. CLIN retrieves successful trials from history.
> > > 2. MemPrompt retrieves the most similar questions with user feedback in the history.
> > > 3. Reflexion utilizes summaries of previous failures.
> > >
> > > In TPD, the student model uses task-level principles from the teacher model in an offline setting. Indeed, TPD can also be augmented by retrieval methods, as the teacher model can provide example-level feedback to the student model. We plan to include related experiments in the revised manuscript.
> > >
> > > **Q6. Section 4.4 page 8 presents an interesting finding that the LLM diverges significantly from its previous answer if a critique or feedback is used. Is this a previously established finding / can you support this claim with any references? This looks contrary to other papers (e.g., cite: [3] page 7) so it is important to substantiate.**
> > >
> > > [6] has discussed some findings about the self-correction capabilities of LLMs. To make this mechanism work, we need external feedback and carefully designed prompts. In our experiments, we provide the principle and ask the student model to critique the previous answer and then revise it based on the critique. We are not sure about the reason, but LLMs are sensitive to prompt formats so we may observe different behaviors when different types of prompts are being used.
> > >
> > >
> > > **Q7. Results are not very convincing: The claim made in the paper are quite broad. However, the empirical results are not too convincing. The improvement over AutoCoT or 3-shot CoT is not significant and it is raises the question whether we can justify a more complex method like yours. It would be helpful to also have a simple baseline that takes the k-examples as demonstrations whose principles are applied in your method.**
> > >
> > > In Figure 6 (in the appendix), we show an ablation study on whether the teacher model needs to provide problem-solving methods in the problem-solving instruction with three examples. We further conducted an experiment on 6-examples selected based on the principle list (there are 6 examples in the final prompt for the student model).
> > >
> > > | GPT-3.5        | Coin  | Letter | Shuffled | Date | Navi. | GSM8K | Matrix | SVAMP |
> > > |----------------|-------|--------|----------|------|-------|-------|--------|-------|
> > > | 6-examples CoT | 92.8  | 83.5   | 75.0     | 74.0 | 93.0  | 74.8  | 89.5   | 82.8  |
> > > | TPD            | 100.0 | 89.9   | 75.0     | 76.5 | 97.5  | 75.4  | 93.5   | 82.9  |
> > >
> > > **Q8. Other Missing details.**
> > >
> > > In the principle exploitation stage, the teacher model modifies the original problem-solving instruction based on the principle list. During the filtering process, we exclude question-incorrect answer pairs that the teacher model cannot accurately distinguish. The true answer is not provided directly to the teacher model. Instead, we first evaluate whether the teacher model can identify errors in the student's answer and then determine if it can generate the correct answer.
> > >
> > > While this filtering process helps ensure quality, the teacher model may occasionally provide vague principles. These instances are flagged for further review and dropped by human reviewers if necessary. At test time, as mentioned in the paper, the teacher model is not involved.
> > >
> > > **Q9. Minor typos: typo in fig 1: promblem (I have not checked more thoroughly for spelling mistakes elsewhere)**
> > >
> > > Thanks for pointing out. We will double-check and modify the writing in the next version.
> > >
> > >
> > >
> > > References:
> > >
> > > [6] Huang, Jie, et al. "Large language models cannot self-correct reasoning yet." arXiv preprint arXiv:2310.01798 (2023).

---

> > > > ### Comment · Reviewer_R4tU · 2024-06-05
> > > > **Strong baseline**
> > > >
> > > > Thank you for the 6-examples CoT -- its avg. performance is 83 across tasks whereas TPD is 86 (with relevant confidence intervals it can be quite close). It reminds of another surprisingly strong baseline that picks 6 most related data points from the training set or the validation set (but not using principles based approach as in TPD). It would be helpful to add this baseline to the next version.
> > > >
> > > > I am wondering about one more perspective to the utility of your work: given as input training examples for a task and a broad description of the task, you can refine the prompt by generating modified problem solving instructions and automatically selecting a set of few shot demonstrations. Is that correct?

---

> > > > > ### Author Response · Authors · 2024-06-06
> > > > > **Discussion with reviewer R4tU**
> > > > >
> > > > > **Q6 Strong baseline that picks 6 most related data points from the training set or the validation set**
> > > > >
> > > > > Thank you for your insightful suggestion. We agree that the baseline of retrieving the six most related data points from the training or validation set is indeed interesting. We share your curiosity about the student model’s performance with this approach. Given the limited time remaining, we will include this baseline in the next version of our paper.
> > > > >
> > > > > **Q7 given as input training examples for a task and a broad description of the task, you can refine the prompt by generating modified problem solving instructions and automatically selecting a set of few shot demonstrations. Is that correct?**
> > > > >
> > > > > Yes, that is correct. With several labels for the questions (used to build practice questions), the teacher model can refine the problem-solving instructions and automatically select a set of few-shot demonstrations. While this paper primarily explores the teacher-student framework, we also envision future work where the student model could self-discover and summarize principles using the true labels from the practice questions.

---

> > > > > > ### Comment · Reviewer_R4tU · 2024-06-06
> > > > > > **Final questions**
> > > > > >
> > > > > > Thank you for the response.
> > > > > >
> > > > > > Ans to Q7  -- future work where the student model could self-discover and summarize principles using the true labels from the practice questions -- at that point, it becomes very close to LEAP [1] method. You can include this in the future work may be.
> > > > > > Ans to Q6 -- strong baseline -- yes, I would be curious to find out in your next version.
> > > > > > Ans to other questions -- thank you. Your responses have given me more clarity now. I will very likely increase my score today. Obviously it would be under the assumption that the authors would include the suggested studies, analysis, baselines, and also try out the method updates (this part, if it is possible to do so). That should make the paper stronger (supporting the claims, and making the method more robust).
> > > > > >
> > > > > > *One final question*: how much human intervention (for quality checks on principles) was needed across different tasks. I am looking for a ballpark quantitative answer. Also, what instruction did you give the human on what a good principle is.

---

> > > > > > > ### Author Response · Authors · 2024-06-06
> > > > > > > **Response to final questions**
> > > > > > >
> > > > > > > We appreciate your thoughtful feedback. Due to the limited time of the discussion period and conference rules, we cannot update the scripts now. However, we guarantee that we will include the suggested studies, analyses, baselines, and method updates in the next version of the paper.
> > > > > > >
> > > > > > > Regarding human intervention for quality checks on principles, since the principle lists in our experiments are not very long, reviewers can finish checking the principle list within approximately 20 minutes per task. In TPD, human reviewers are responsible for verifying each principle in the principle list for all tasks to ensure quality. They sequentially review and remove any principles that are vague, erroneous, or confusing.
> > > > > > >
> > > > > > > The instructions provided to human reviewers are as follows. We will also include them in the appendix in the next version.
> > > > > > >
> > > > > > > 1.The principle should be written in simple, clear language that is easy to understand.
> > > > > > >
> > > > > > > 2.The principle should be specific enough to address a particular type of error or concept.
> > > > > > >
> > > > > > > 3.The principle should be factually correct and free from errors.
> > > > > > >
> > > > > > > 4.The principle should provide actionable guidance that helps the student avoid the error in the future.
> > > > > > >
> > > > > > > 5.While specific, the principle should also be general enough to apply to similar errors or situations.

---

> > > > > > > > ### Comment · Reviewer_R4tU · 2024-06-06
> > > > > > > > **Thank you.**
> > > > > > > >
> > > > > > > > Thank you. I increased my score accordingly.

---

> > ### Comment · Reviewer_R4tU · 2024-06-05
> > **Thank you for the detailed response.**
> >
> > Dear authors,
> >
> > I needed some clarifications based on the rebuttal content you wrote (which was quite helpful, thank you):
> >
> > - line 7 of algorithm 1 where `P += p` , we do not check for near-duplicates. Does it happen in practice that P has many near-duplicates? If P indeed has near-duplicates, then the selected examples might be over-represented because of the near-duplicates. What mechanism prevents from happening? Is it true that line 7 of the algorithm should really also have a call to a human: `P += p  if is_good_principle(p) else empty`. Related: what exactly is a good principle-- one that is crucial e.g., it is the cause of most erroneous examples? I am assuming there is no _weight_ given to a principle currently
> >
> > - Thank you for the table on the influence of the size of the validation set on TPD’s performance. I could not find this in the paper: please include this important information in the next version.
> >
> > - _Our TPD method selects examples based on the number of violations against the principle list. However, we can modify it to ensure higher coverage of the principle list, further optimizing example selection._ Yes that would help. How do you plan to include this modification-- do you plan to formulate it as an optimization problem: given N principles,  and each principle having a weight w, select k principles such that they are diverse, crucial, correct.
> >
> > - for open ended tasks: _we can choose examples based on questions that are prone to violating the principles on different desirable aspects of the task._ that is good, and does it mean that these aspects essentially become principles and you will supply them manually and not discover them -- would that simplify the current setup?

---

> > > ### Author Response · Authors · 2024-06-06
> > > **Discussion with Reviewer R4tU**
> > >
> > > We appreciate the reviewers' insightful feedback.
> > >
> > > **Q1 Do near-duplicates happen in practice that P has many near-duplicates? what exactly is a good principle?**
> > >
> > > Thank you for raising this question! In practice, we do not find near-duplicate principles. In TPD, we check the existing principle list P in line 5 of Algorithm 1 to prevent this issue. Intuitively, if the existing principle list P cannot address the presented error, the teacher model will formulate a new principle p for it. We agree that a good principle should cover a type of error. We acknowledge that we do not assign weights to principles at this stage. If the principle list becomes longer and longer, we can attempt to ask the teacher models to merge some principles. For example, if two principles address the problem of the same type, from different perspectives, the teacher model could merge them.
> > >
> > > **Q2 the influence of the size of the validation set on TPD’s performance**
> > >
> > > Thanks for the suggestion!  Although the conference guidelines prevent us from updating the paper at this stage, we will ensure that this important information is included in the next version of the paper.
> > >
> > > **Q3 How do you plan to include this modification-- do you plan to formulate it as an optimization problem?**
> > >
> > > Thank you for your insightful suggestion. Yes, we plan to formulate this as an optimization problem, as you described. After constructing the principle list P, we will have the teacher model revisit it and assign a weight to each principle based on its importance to the question and its similarity to other principles. We will then define the violation score as the summation of the weights of the violated principles for an incorrect question-pair in the practice questions. Based on this, we will select k examples that cover diverse, crucial, and correct principles.
> > >
> > > **Q4 for open ended tasks:**
> > >
> > > For open-ended generations, there are some common important aspects, such as helpfulness and harmlessness, which we can use as predefined principles. However, these manually defined principles may not fully cover all desired aspects in the generated outputs. We may still need to discover some model-specific principles through the error summarization stage. This will depend on the specific student models, and we need to conduct empirical studies to investigate this further.

---

> > ### Comment · Reviewer_R4tU · 2024-06-05
> > **Error analysis**
> >
> > Dear authors,
> >
> > Thank you for acknowledging the feedback on adding a retrieval module and augmenting with feedback; and for including an error analysis subsection. That would make the setup more scalable and robust.
> >
> > - Your error analysis needs to be more thorough. E.g., you claim that "The teacher model is effective in identifying the student's failures, showing a relatively low number of failures" but the paper [6] that you cited and also [7] has evidence to suggest that for some reasoning tasks such as in math reasoning even larger models cannot spot a mistake (e.g, if the answer has a CoT style answer). The analysis that some principles are unhelpful is useful -- please include a more thorough version of this in the next version of your paper, clearing highlighting what kind of problem contributes to roughly what % of error with some examples of each error type.
> >
> > [6] Huang, Jie, et al. "Large language models cannot self-correct reasoning yet." arXiv preprint arXiv:2310.01798 (2023).
> > [7] Madaan et. al, "Self-Refine: Iterative Refinement with Self-Feedback" NeurIPS 2023

---

> > > ### Author Response · Authors · 2024-06-06
> > >
> > > **Q5 Your error analysis needs to be more thorough**
> > >
> > > We acknowledge that the teacher model is generally effective at identifying the student's failures. However, there are some complex reasoning tasks, where instances fail to detect errors, even with larger models. This observation underscores the necessity of the filtering module in TPD where we filter out undetected errors to enable the teacher model to summarize principles from the errors it can identify.
> > >
> > > When we describe the teacher model as "effective," we refer to its ability to identify a relatively low number of failures compared to its successes. However, we recognize the need for a more thorough analysis. In the next version of our paper, we will conduct further studies to categorize the types of problems contributing to errors and quantify the percentage each type represents. Additionally, we will provide examples of each error type to enhance understanding and clarity.

---

### Decision · Program_Chairs · 2024-07-10

**Decision:**

Accept

**Comment:**

The paper introduces a method for prompting student models based on error analysis and guidance from a larger model. Evalauting this approach on 2 LLMs with a number of tasks demonstrates higher performance compared to CoT variants. Ablations demonstrate that reasonable performance can be obtained with few shot settings which will reduce cost of inference. All high quality reviews assign a score of marginal accept and the reviewer score improved during the rebuttal phase: this introduced several new baselines and ablations which will result in a stronger paper. There are several common weaknesses identified by the reviewers, particularly around the failure cases and handling factual knowledge. I look forward to seeing an updated version of the paper or future work which would discuss these issues.

[At least one review was discounted during the decision process due to quality]